biomechanics

fish swimming, optimization strategy, cost of transport, swimming energetics, tail-beat frequency, drag

**Authors for correspondence:**
Gen Li
e-mail: ligen@jamstec.go.jp
Hao Liu
e-mail: hliu@faculty.chiba-u.jp
Johan L. van Leeuwen
e-mail: johan.vanleeuwen@wur.nl

# Fishes regulate tail-beat kinematics to minimize speed-specific cost of transport

Gen Li[1], Hao Liu[2], Ulrike K. Müller[3], Cees J. Voesenek[4] and Johan L. van Leeuwen[4]

[1]Center for Mathematical Science and Advanced Technology, Japan Agency for Marine-Earth Science and Technology (JAMSTEC), 3173-25, Showa-machi, Kanazawa-ku, Yokohama, Japan
[2]Graduate School of Engineering, Chiba University, 1-33, Yayoi-cho, Inage-ku, Chiba, Japan
[3]Department of Biology, California State University, Fresno 2555 E San Ramon Avenue, Fresno, CA 93740, USA
[4]Experimental Zoology Group, Department of Animal Sciences, Wageningen University, De Elst 1, 6708 WD, Wageningen, The Netherlands

 GL, 0000-0002-4423-3266; HL, 0000-0002-8687-3237; CJV, 0000-0002-5467-8963; JLvL, 0000-0002-4433-880X

Energetic expenditure is an important factor in animal locomotion. Here we test the hypothesis that fishes control tail-beat kinematics to optimize energetic expenditure during undulatory swimming. We focus on two energetic indices used in swimming hydrodynamics, cost of transport and Froude efficiency. To rule out one index in favour of another, we use computational-fluid dynamics models to compare experimentally observed fish kinematics with predicted performance landscapes and identify energy-optimized kinematics for a carangiform swimmer, an anguilliform swimmer and larval fishes. By locating the areas in the predicted performance landscapes that are occupied by actual fishes, we found that fishes use combinations of tail-beat frequency and amplitude that minimize cost of transport. This energy-optimizing strategy also explains why fishes increase frequency rather than amplitude to swim faster, and why fishes swim within a narrow range of Strouhal numbers. By quantifying how undulatory-wave kinematics affect thrust, drag, and power, we explain why amplitude and frequency are not equivalent in speed control, and why Froude efficiency is not a reliable energetic indicator. These insights may inspire future research in aquatic organisms and bioinspired robotics using undulatory propulsion.

## 1. Introduction

Undulatory swimming is the most common swimming style in fishes across a wide range of body sizes and shapes [1–5]. Undulatory swimmers generate transverse bending waves along the body that may be visible along their posterior (carangiform swimmers) or entire body (anguilliform swimmers) [2]. Swimmers can control swimming speed by altering body-wave frequency and tail-beat amplitude, according to mathematical models [6–8]. Yet fishes appear to favour varying frequency [7], according to the observation that swimming speed is proportional to tail-beat frequency $f$ over a wide range of body kinematics, geometry, and size (figure 1a) whereas tail-beat amplitude $A$ (expressed in units of body length) varies much less and non-linearly across swimming speeds (figure 1b).

With energetics being an important factor during routine activities, fishes might change frequency rather than amplitude to optimize energetic expenditure during cyclic swimming. To study fluid dynamic energetic expenditure, scientists developed two indices [8], the dimensionless Froude efficiency $\eta$ [4,6] and cost of transport $\Omega$ [12–15]:

$$\eta = \frac{TU}{P} \tag{1.1}$$

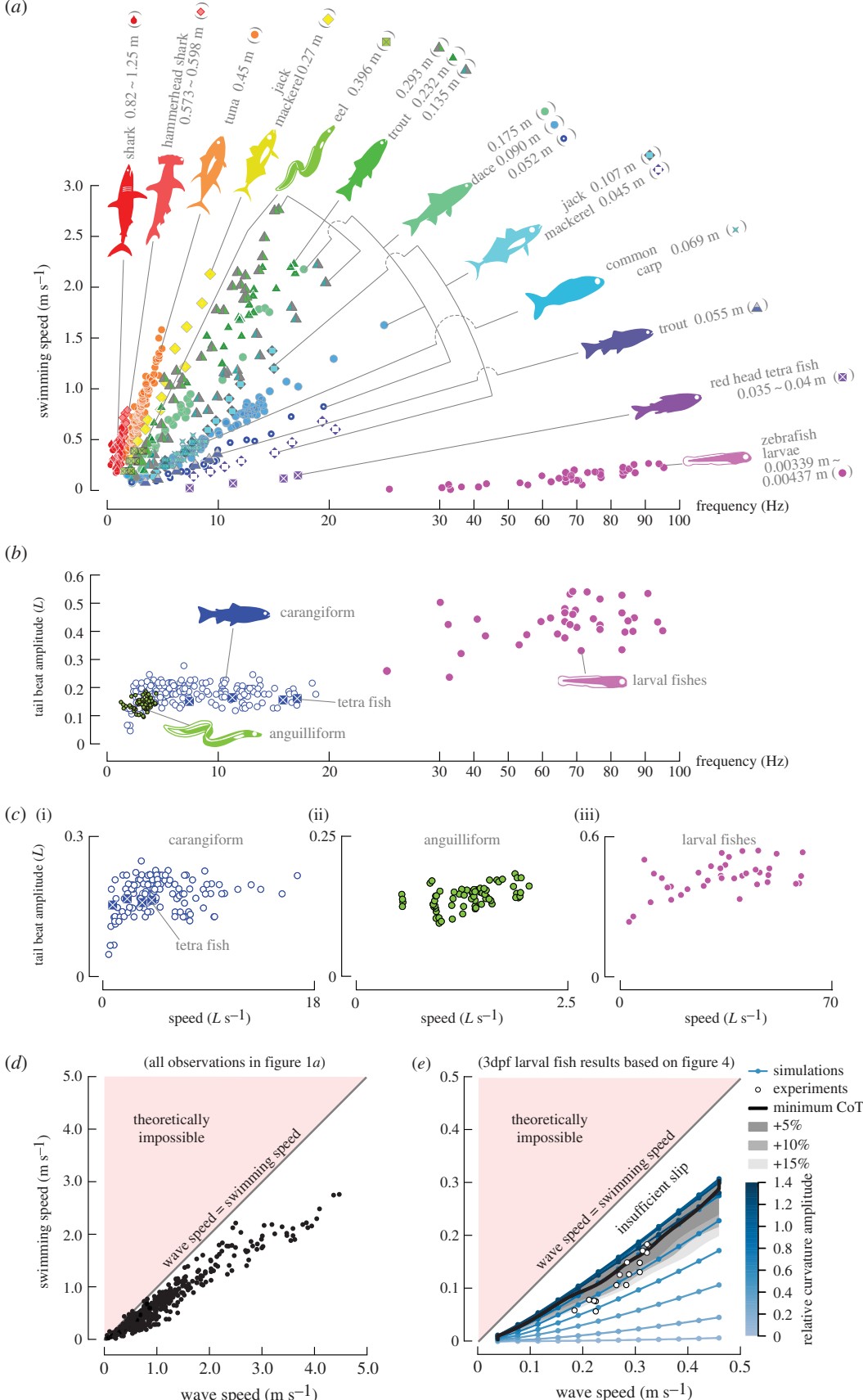

**Figure 1.** Swimming speeds tends to scale linearly with tail-beat frequency and body-wave speed, but not amplitude. (a) Experimental observations of frequency-speed relationships of a variety of fishes and body lengths (sources: see the electronic supplementary material table S5). (b) Normalized peak-to-peak amplitude–frequency relationships in carangiforms (trout and dace, [7]; tetra fish symbol '⊠', [9]), anguilliforms (eel, [10]) and larval fishes (zebrafish, [11]). (c) Normalized peak-to-peak amplitude–speed relationships (sources: same as panel (b)). (d) Data points in panel (a) collapse into one data cloud when plotting swimming speed against body-wave speed $w$ ($w = fL$). (e) Relationship between wave speed and swimming speed for 3dpf larval fish results. Experimental $w$ values (white circles: identical to observations in figure 4b) cluster at or just below the predicted speed-specific cost-of-transport (CoT) optimum (black line) and occupy only a narrow region in the parameter space of theoretically possible amplitude–frequency combinations (blue dots). (Online version in colour.)

and

$$\Omega = \frac{E}{Sm} = \frac{P}{Um},$$ (1.2)

where $T$, $U$ and $P$ are thrust, swimming speed and input power, respectively (each averaged over one tail beat cycle), $E$ is energy consumption, $S$ is displacement and $m$ is body mass. The two indices differ in which aspect of hydrodynamic performance is optimized and consequently are likely to peak at different swimming kinematics and speeds. Optimizing $\eta$ requires maximizing the ratio of useful power to power expended to the water, irrespective of net energetic expenditure per unit distance. By contrast, optimizing $\Omega$ requires minimizing input power to move a given mass at a given speed. Previous studies proposed that fishes swim near a particular Strouhal number ($Af/U$) to maximize $\eta$ [16] at a given speed, but other studies question this explanation [8,13].

Here, we examine if fishes regulate tail-beat kinematics to optimize energetic expenditure during undulatory swimming. To test this hypothesis, we ask two questions: (i) do body-wave frequency and swimming speed correlate owing to energetic expenditure optimization? and (ii) if optimizing energetic expenditure at a given speed, do fishes minimize cost of transport or maximize Froude efficiency? To answer these questions, we combined numerical modelling and experimental observations. We used computational fluid dynamics (CFD) to simulate experimental outcomes and computed counterfactual cases to describe the performance surface of fishes as a function of body-wave frequency and amplitude for two mathematically modelled body-wave types (carangiform, anguilliform) and one experimentally observed body wave (based on a larval zebrafish) (figures 2a, 3a and 4a). Combining these high-resolution performance maps with our extensive experimental dataset on larval fishes allowed us to go beyond previous numerical studies [17,18] and actually test hypotheses about optimization strategies used by actual fishes. We show that fishes can control swimming speed by changing frequency rather than amplitude to minimize fluid-dynamic speed-specific cost of transport rather than maximize Froude efficiency, and that this strategy successfully predicts experimentally observed behaviours in both the laminar and turbulent flow regime.

## 2. Material and methods

### (a) Experimental data

We used experimental data from published studies (figure 1; sources in electronic supplementary material, §F) and new experiments (figure 4). We recorded cyclic swimming episodes of larval zebrafish at age 3 days post fertilization (dpf) using a set-up with three synchronized high-speed cameras described in more detail in previous publications [19,20]. The experimental set-up is described in the electronic supplementary material, §E1.

### (b) Numerical methods

We developed a three-dimensional Navier–Stokes solver based on a finite volume method [21,22] that has been validated and used to model swimming and flying animals [15,23]. By coupling hydrodynamic and body-dynamic solutions (see the electronic supplementary material, §B), this numerical approach inputs the swimmer's body-shape changes (internal kinematics) to output external kinematics (centre-of-mass movements and body orientation). The CFD model comprises a surface model of the changing fish shape, and a local fine-scale body-fitted grid plus a stationary global grid to compute the flow patterns around the fish with sufficient resolution. Navier–Stokes equations were solved in each grid and results were interpolated at the grid interfaces.

### (c) Morphology and kinematic modelling

We simulated a carangiform swimmer with a body shape approximating an adult tetra fish, an anguilliform swimmer with the body shape approximating a juvenile eel, and a swimmer with the body of a larval zebrafish using shape and size measurements from a larva aged 3 dpf (figures 2a, 3a and 4a). The body length of the carangiform swimmer and the juvenile anguilliform swimmer were set to 20 mm and their body waves were prescribed by sinusoidal functions:

$$\begin{cases} H(l, t) = \alpha \cdot l^2 \cdot \sin\left(\frac{2\pi l}{\lambda} - 2\pi ft\right) & \text{(carangiform swimmer)} \\ H(l, t) = \alpha \cdot e^{l-1} \cdot \sin\left(\frac{2\pi l}{\lambda} - 2\pi ft\right) & \text{(anguilliform swimmer)}, \end{cases}$$ (2.1)

where $\alpha$ is the amplitude control factor, $l \in [0, 1]$ is the dimensionless distance from the snout along the longitudinal axis, $H(l, t)$ is the dimensionless lateral excursion at time $t$, $\lambda$ is the dimensionless length of the body wave based on body length (carangiform: $\lambda = 1.1$; anguilliform: $\lambda = 0.64$ [24,25]).

The body wave and size of the larval fish is based on an experimental observation of a larval zebrafish aged 3 dpf (see the electronic supplementary material, §E2). We defined the instantaneous body axial line as:

$$X(l, t) = \int_0^l \beta \cdot c\left(l, \frac{f}{f_p}t\right) \, dl,$$ (2.2)

where $t$ is the time in the simulation, $f_p$ is the frequency observed in the experiment, $\beta$ and $f$ are respectively body curvature control factor and tail-beat frequency. $c(l, t)$ is a periodic function with respect to time $t$ and repeating on intervals of 1 (i.e. $c(l, t+1) = c(l, t)$), which represents the curvature time series observed in the experiment. $l \in [0, 1]$ is the dimensionless distance from the snout along the longitudinal axis of the fish.

We simulated a wide range of tail-beat frequency and amplitude combinations, including counterfactual ones (white dots in figures 2c, 3c and 4c), to interpolate a parameter space relevant for our optimization study.

### (d) Validation of numerical methods

We limited Reynolds numbers ($Re$) in this study to values from 1 to 6000 by capping the body length of swimmers at 20 mm. This $Re$ range allowed us to perform a large number of simulations at a previously validated grid resolution [26] with feasible time cost and without requiring a turbulence model, affording us high accuracy while maintaining key flow features that are robust against a reduction in $Re$ [27]. We validated our simulations against experimental observations (figure 4): (i) our simulations predict the swimming speed of the reference case (figure 4c, red dot) to within 1%; and (ii) predicted and observed speed are in good agreement for all experimental data points (figure 4b).

### (e) Measurements of speeds, forces and energetics

In each simulation, the model fishes accelerated from rest until thrust matches drag, resulting in an asymptotic increase in cycle-averaged swimming speed. We defined cyclic swimming as cycle-averaged swimming speed increasing by less than 1% from the previous cycle. All CFD results reported in this study were computed after the swimmer reached cyclic swimming.

Instantaneous thrust (drag) at each time-step was defined as the sum of the forward (backward) components of pressure and

*Proc. R. Soc. B* **288**: 20211601

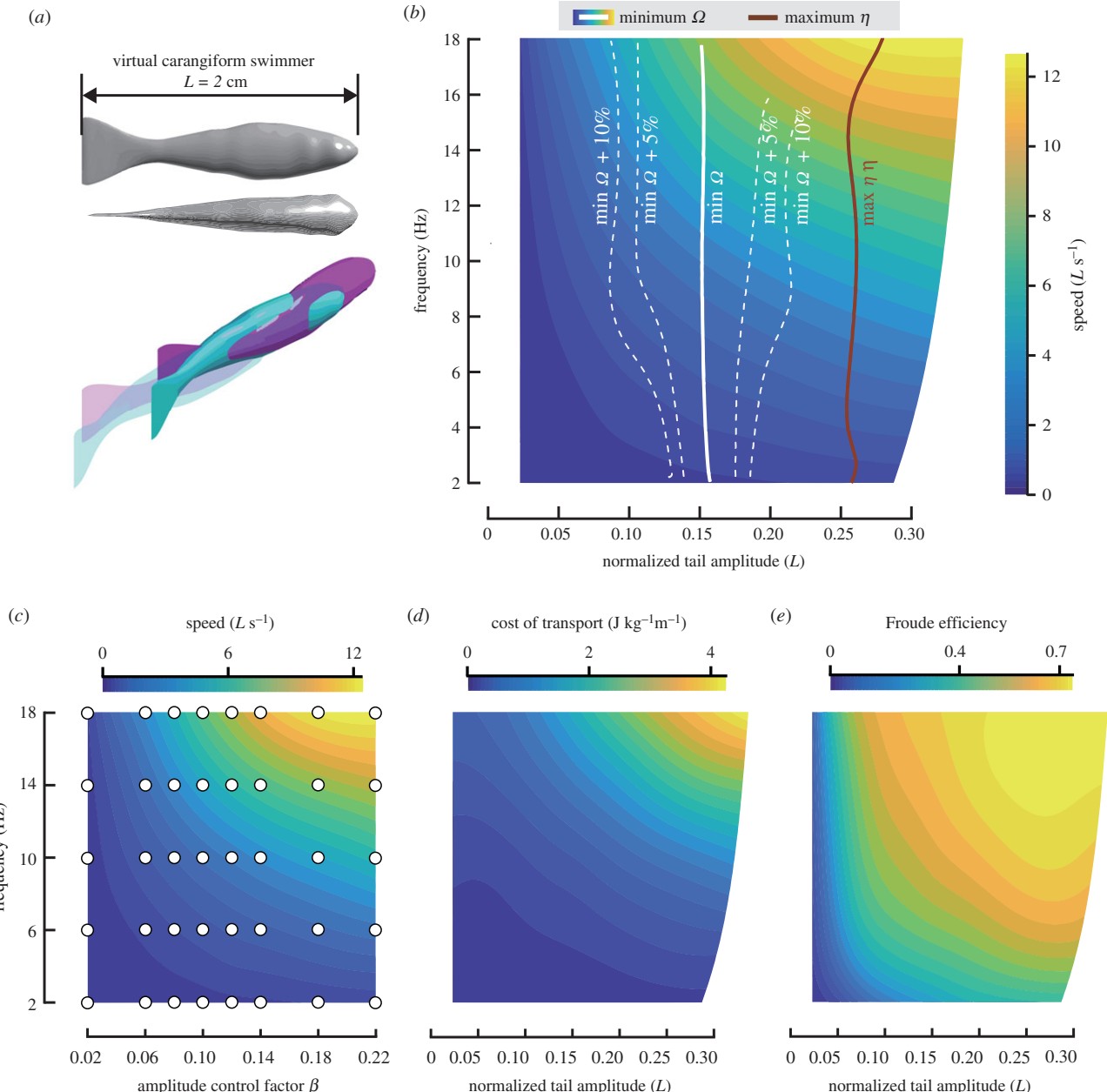

**Figure 2.** Performance maps for a simulated carangiform swimmer: heatmap of swimming speed as a function of body wave frequency and amplitude. (*a*) Swimmer's body shape and motion. (*b*) Speed-specific minimum-cost-of-transport ($\Omega$) curve (white line; white dashed line: +5% and +10%) and maximum-Froude-efficiency ($\eta$) curve (brown line), superimposed on heat map. (*c*) Distribution of CFD simulations across the performance map: heatmap of swimming speed as a function of frequency and tail-beat amplitude control factor, based on 40 simulations (white dots). (*d*) Cost of transport ($\Omega$), and (*e*) Froude efficiency ($\eta$). Note that in (*b*, *d* and *e*) the horizontal axis denotes normalized tail-beat amplitude instead of the tail-beat amplitude control factor in (*c*). Owing to the nonlinear relationship between tail-beat amplitude and tail-beat amplitude control factor, the heat map is no longer rectangular (electronic supplementary material, figures S10 and S11). (Online version in colour.)

shear stress over all fish surface elements (electronic supplementary material, figure S4). At any instant, a surface element may contribute thrust or drag, and this contribution is owing to pressure and shear stress acting on the element. Total thrust and drag were computed by averaging over one tail-beat cycle. We use power to refer to mechanical power, defined as the sum of hydrodynamic and body inertial powers. Hydrodynamic power was calculated as the sum of the hydrodynamic work per unit time on the body surface. Body inertial power was computed as the sum of the kinetic energy change rate of all body elements. Froude efficiency $\eta$ was calculated by equation (1.1), cost of transport $\Omega$ by equation (1.2).

We examined two optimization strategies to find optimal combinations of $A$ and $f$ as a function of swimming speed $U$: one for minimizing cost of transport $\Omega$ and another for

maximizing Froude efficiency $\eta$ (see the electronic supplementary material, figure S9 for a detailed explanation on how we found optimal trajectories).

## 3. Results

### (a) Energetic optimization by carangiform and anguilliform swimmers

First, we studied a carangiform swimmer and an anguilliform swimmer, building amplitude–frequency parameter space maps (figure 2*b*,*d* and *e*; figure 3*b*, *d* and *e*) through interpolation between simulated cases (white points, figure 2*c* and

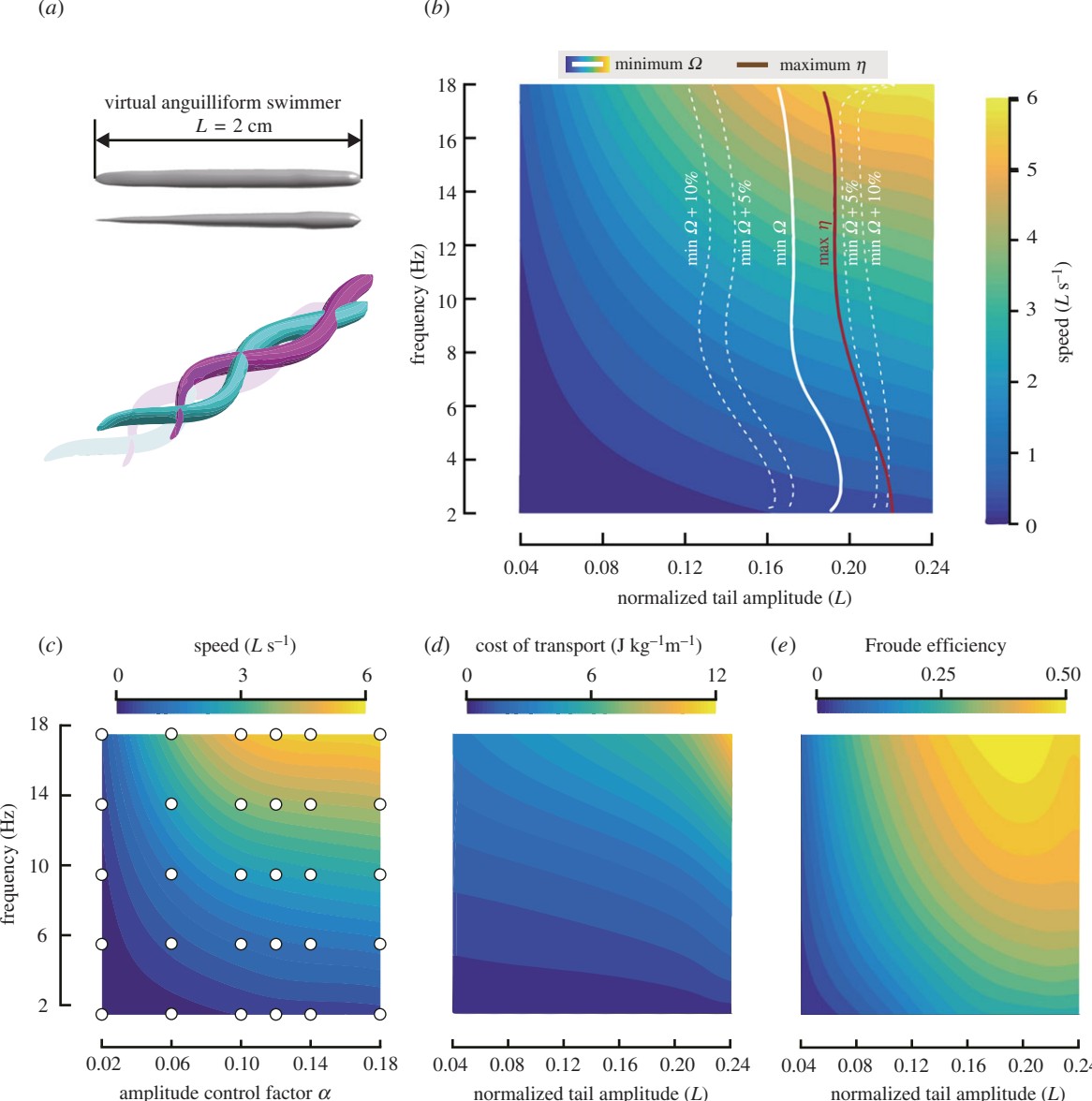

**Figure 3.** Performance maps for a simulated anguilliform swimmer: heatmap of swimming speed as a function of body wave frequency and amplitude. (*a*) Swimmer's body shape and motion. (*b*) Speed-specific minimum-cost-of-transport ($\Omega$) curve (white line; grey dashed lines: +5%, +10%) and maximum-Froude-efficiency ($\eta$) curve (brown line), superimposed on the heat map. (*c*) Distribution of CFD simulations across the performance map: heatmap of swimming speed as a function of frequency and tail-beat amplitude control factor, based on 30 simulations (white dots). (*d*) Cost of transport ($\Omega$), and (*e*) Froude efficiency ($\eta$). Note that in (*b, d* and *e*) the horizontal axis denotes normalized tail-beat amplitude instead of the tail-beat amplitude control factor in (*c*). (Online version in colour.)

figure 3*c*). We found that the two swimmers can swim faster by either increasing tail-beat frequency or amplitude (figure 2*b* and figure 3*b*). For both swimmers, the $\Omega_{min}$ strategy predicts a nearly vertical optimality trajectory, suggesting that fishes should change frequency to control speed, while keeping amplitude within a narrow range (carangiform: 0.14–0.16 *L*; anguilliform: 0.18 *L*; *L* is body length) (figures 2*b*, 3*b*). The $\eta_{max}$ strategy predicts a similarly vertical trajectory but further to the right in the parameter space, suggesting that the $\eta_{max}$ strategy requires larger tail-beat amplitudes at all speeds than the $\Omega_{min}$ strategy.

To explore which optimization strategy comes closest to what actual fishes do, we compared our predictions with published experimental data (references in the electronic supplementary material, §F). Amplitude values for dace (0.17 *L*; 0.153 *L*), trout (0.17 *L*) and tetra fish (0.16 *L*) are close to or within the predicted optimal range for $\Omega_{min}$ (0.14–016 *L*) (figure 1*b,c*) but not for $\eta_{max}$ (greater than 0.25 *L*) (figure 2*b*).

The same is true for anguilliform swimmers, with experimentally observed values in eel (0.11–0.18 *L*) closer to the predicted $\Omega_{min}$ (figure 1*b,c*) than the predicted $\eta_{max}$ strategy (figure 3*b*). The eel data show a relatively low swimming speed and narrow speed range, which ensure an overall low cost of transport.

In conclusion, comparing the two predicted optimization trajectories through the parameter space with experiments suggests that fishes minimize $\Omega$ rather than maximize $\eta$. Furthermore, fishes should increase speed by increasing tail-beat frequency while keeping tail-beat amplitude approximately constant to stay close to the speed-specific $\Omega_{min}$.

## (b) Energetic optimization by larval fishes

To verify our simulations, we modelled a larval zebrafish (age 3 dpf), which naturally swims in the laminar flow regime and for which we have detailed morphological and experimental data.

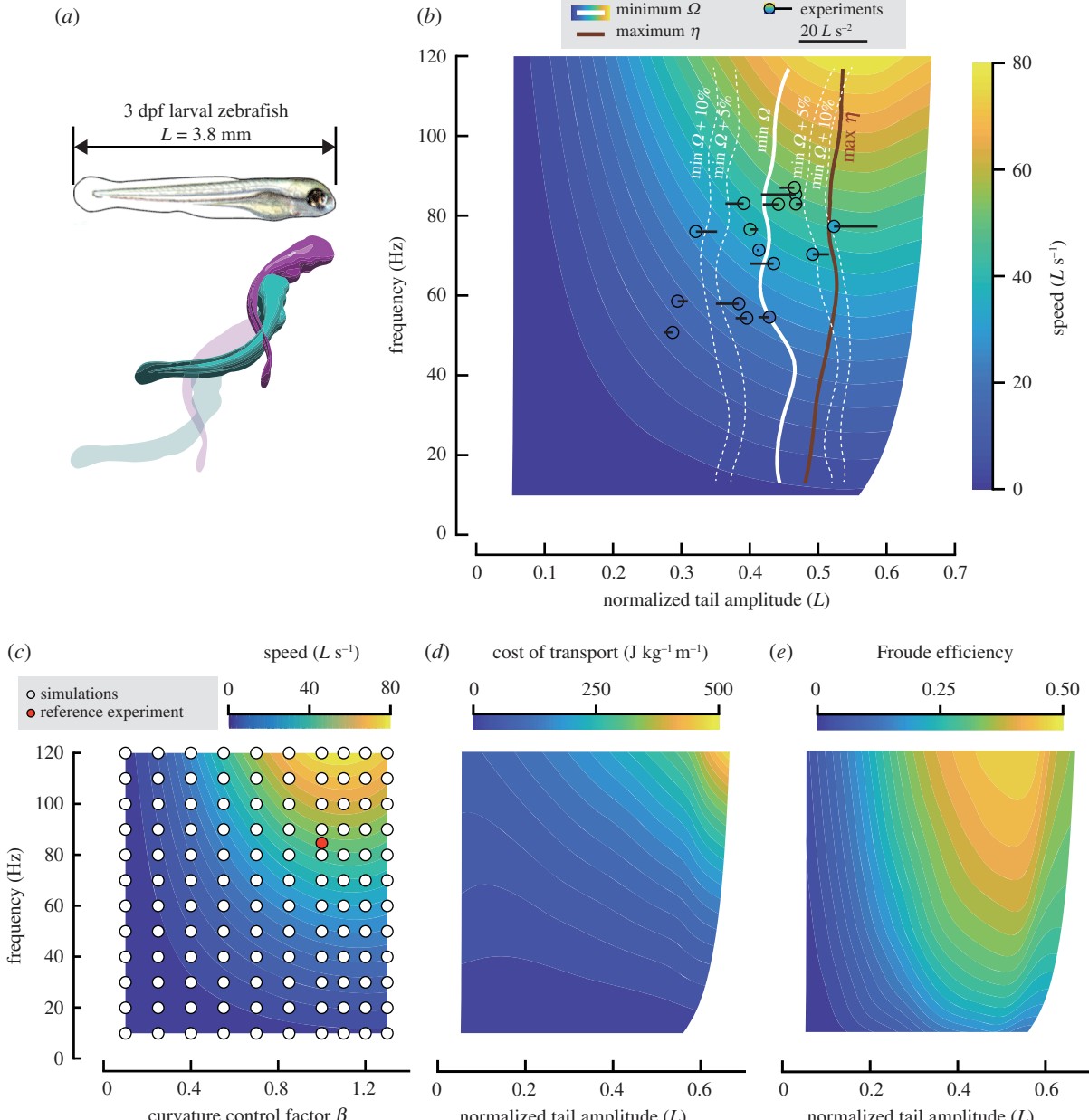

**Figure 4.** Performance maps for a larval fish. (*a*) Larval body shape and swimming motion. (*b*) Observed kinematics of fish larvae is closer to the minimum-speed-specific-cost-of-transport ($\Omega$) curve (white line; grey dashed lines: +5%, +10%) than to the maximum-speed-specific-Froude-efficiency curve (brown line); black circles: experimental observations, fill colour of black circles: observed swimming speed (validation: good agreement with model prediction), black line originating in each circle: rate of change in swimming speed, line to left: deceleration, line to right: acceleration. (*c*) Distribution of CFD simulations across the performance map: heatmap of swimming speed as a function of frequency and body-curvature control factor, based on 120 simulations (white dots); red dot: reference case (model validation: predicted speed = 1.01 observed speed). (*d*) Cost of transport ($\Omega$), and (*e*) Froude efficiency ($\eta$). Note that in (*b, d* and *e*) the horizontal axis denotes normalized tail-beat amplitude to facilitate comparison with experimental data. Owing to the non-linear relationship between tail-beat amplitude and body-curvature control factor, the heat map is no longer rectangular (electronic supplementary material figures S10 and S11). (Online version in colour.)

The maps of performance indicators $U$, $\Omega$, and $\eta$ as functions of $f$ and $A$ (figure 4*b,d,e*) show a similar trend as the simulated carangiform and anguilliform swimmers. Swimming speed increases with $f$, irrespective of $A$ (figure 4*b*). Swimming speed increases with $A$, except for values above 0.55. The predicted $\Omega_{min}$ and $\eta_{max}$ trajectories do not coincide, and both show a trend predominantly in the frequency direction, making frequency the main parameter to control speed for both strategies. Along the $\Omega_{min}$ and $\eta_{max}$ trajectories, $A$ varies slightly and remains close to 0.45 $L$ and 0.55 $L$, respectively.

Comparing the two predicted trajectories with experimental data suggests that fish larvae minimize $\Omega$ rather than maximize $\eta$ (figure 4*b*): experimental data cluster around the $\Omega_{min}$ trajectory (75% of the data are within the 5% interval of the $\Omega_{min}$ trajectory; white dotted curves in figure 4*b*). Fish larvae following a $\Omega_{min}$ strategy should raise tail-beat frequency to increase swimming speed, while keeping tail-beat amplitude nearly constant. This implies that speed during (nearly) cyclic swimming should correlate strongly with frequency. We selected experimentally observed near-cyclic swimming events with small accelerations and decelerations (indicated by short horizontal bars in figure 4*b*). During linear accelerations (decelerations), larval fishes tend to increase (decrease) tail-beat amplitude as apparent from two data points at the right-hand side in figure 4*b*, which lie closer to the $\eta_{max}$ trajectory than the

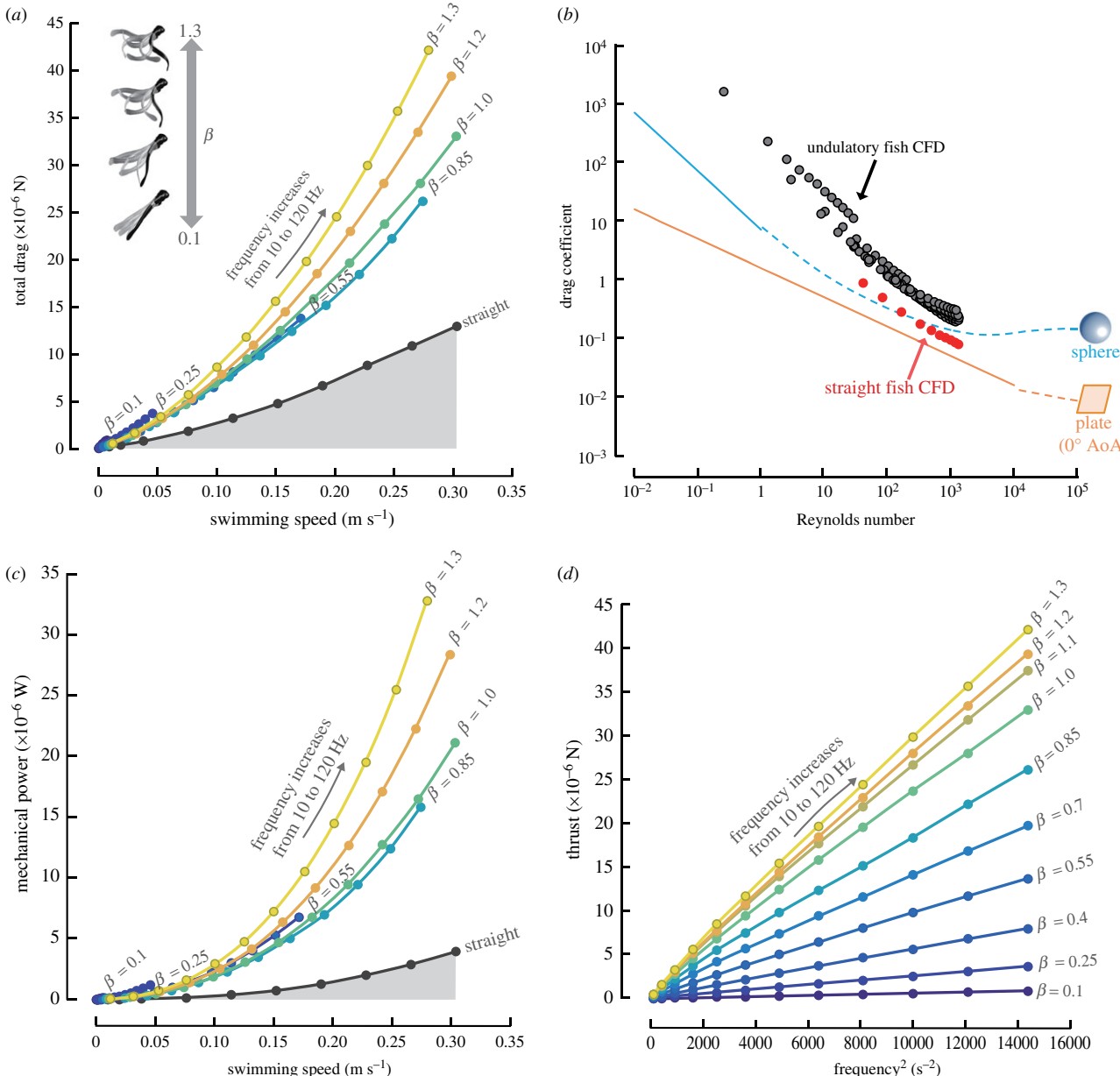

**Figure 5.** Computed drag, thrust and fluid dynamic power generated by a swimming larval fish. (a) Total drag on the undulatory swimming fish at various tail-beat amplitudes (indicated by body curvature control factor $\beta$), as well as a straight fish, covering the same speed range. (b) Computed drag in undulatory swimming fish; CFD simulations for fish undulating their body (grey dots) and fish gliding with a straight body (red dots). Blue lines: predictions of three-dimensional analytical models for a sphere in the viscous ($C_D = 6/Re$, solid line), transitional, and inertial flow regimes ($C_D = 6(1 + 0.15Re^{0.687})/Re + 0.105/(1 + 42500Re^{-1.16})$, dashed line) based on wetted area [28]; orange line: predictions of a two-dimensional analytical model for a plate with zero angle of attack, in the laminar ($C_D = 1.33Re^{-0.5}$, solid line) and turbulent ($C_D = 0.072Re^{-0.2}$, dashed line) regimes [1]. (c) Predicted mechanical power of the undulatory swimming fish at various $\beta$, as well as a straight fish model. (d) Thrust consistently increases as curvature factor $\beta$ increases, and at a given $\beta$, thrust is roughly proportional to $f^2$. (Online version in colour.)

$\Omega_{min}$ trajectory. The slowest swimming events deviate most from the predicted $\Omega_{min}$ trajectory because a close match may not be critical when energetic expenditure is very low.

In conclusion, we can now answer both research questions of this study: undulatory swimmers control their speed by changing tail-beat frequency, which allows them to minimize speed-specific cost of transport rather than maximize speed-specific Froude efficiency.

## (c) Undulation affects drag and thrust

To determine the hydrodynamic mechanism that places the two optimization strategies on different trajectories through the parameter space, we examined how body-wave kinematics affect thrust, drag and power. By computing total

drag acting on a fish larva for many combinations of frequency $f$ and curvature control factor $\beta$, we found that total drag on an undulating fish exceeds drag on a straight fish moving at the same constant speed by a factor 2–3 (figure 5b). Both pressure- and viscous-drag components increase owing to body undulation (electronic supplementary material, figure S12). Low body curvatures lead to low swimming speeds, and drag is relatively large at a given speed (e.g. curves with $\beta = 0.1$ and 0.25 in figure 5a); a large $\beta$ also results in a relatively large drag (e.g. curves with $\beta = 1.2$ and 1.3 in figure 5a) and thrust (figure 5d). For the highest frequency, increasing $\beta$ above about 1.1 increases thrust and drag, but they balance at a decreasing maximum speed while power input increases (figure 5c), making this $\beta$-increase ineffective. The optimal amplitude predicted by the $\Omega_{min}$

trajectory (figure 4b) corresponds to a β-value for minimum drag (0.85–1.00).

Comparisons with a sphere and a flat plate reveal that the drag coefficient $C_D$ (based on wetted surface) of an undulating fish scales with $Re$ in a similar way as other three-dimensional objects (sphere, coasting fish), but drag varies as the body undulates and is much higher in magnitude. Importantly, the drag on an undulating fish is not predicted well by two-dimensional analytical models (i.e. a flat plate with zero angle of attack) that ignore how three-dimensional flow and body undulations affect drag, capturing neither its magnitude nor its scaling with $Re$ (figure 5b) [6,15,29,30]. At intermediate $Re$, CFD predictions and experimental observations of larval fishes deviate from scaling relationships predicted by two-dimensional models [31]. In the low $Re$ viscous flow regime, drag scaling with $Re$ is predicted well by the three-dimensional Stokes flow around a sphere ($C_d \propto Re^{-1}$), but not by the two-dimensional Blasius flow around a flat plate with zero angle of attack ($C_D \propto Re^{-1/2}$).

We computed the power needed to drive a straight fish body forward at constant speed $U$, by $P_{straight} = U \cdot D_{straight}$, where $D_{straight}$ is the drag on the straight body (figure 5c). The power $P_{undulation}$ produced by an undulating fish exceeds that of a straight fish at the same speed by a factor of 5 up to greater than 8. Very small and very large amplitudes require more power than the optimal amplitude corresponding to the $\Omega_{min}$ trajectory in figure 4b.

In answer to the question of why the optimal trajectory of the $\Omega_{min}$ strategy requires lower tail-beat amplitudes than the $\eta_{max}$ strategy, we found that the $\Omega_{min}$ strategy constrains amplitude whereas the $\eta_{max}$ strategy does not: drag on an undulatory swimmer depends not solely on swimming speed, but also the kinematics of the undulating body; hence the swimmer needs to optimize its body kinematics to prevent excessive energy consumption by limiting drag according to the $\Omega_{min}$ strategy.

### (d) Swimming speed and tail-beat frequency correlate owing to energetic optimization, not fluid-dynamic constraints

To test if the tight correlation between swimming speed and tail-beat frequency observed in fishes results from energetic optimization or hydrodynamic constraints, we compare a fish's theoretically possible parameter space with their inhabited performance space. We estimate the theoretically possible space by varying body kinematics (frequency $f$, curvature index $\beta$) within biologically relevant limits.

We know that thrust $T$ increases with curvature control factor $\beta$, irrespectively of $f$. At a given $\beta$, $T$ is approximately proportional to $f^2$, in particular at high swimming speeds (figure 5d). As $Re$ increases from $10^0$ to $10^3$, the relationship between body drag and speed changes from $D \propto U$ (i.e. $C_D \propto Re^{-1}$ at $Re \sim O$ (less than $10^0$)) to $D \propto U^2$ (i.e. $C_D \rightarrow$ const. at $Re \sim O$ (greater than $10^3$)) (figure 5b). Because time-averaged drag and thrust match during cyclic swimming, and thrust is approximately proportional to $f^2$ ($T \propto f^2$) and $D \propto U^2$ at $Re \sim O$ (greater than $10^3$), the relationship between $f$ and $U$ changes to $U \propto f$ at high speeds. However, such fluid-dynamic relationships are attached to a specific curvature control factor $\beta$, a controller of tail-beat amplitude. If fishes changed their tail-beat amplitude freely, there would

be no tight linear correlation between swimming speed and tail-beat frequency.

Energetic optimization requires a linear correlation between swimming speed and tail-beat frequency. The $\Omega_{min}$ strategy predicts that tail-beat amplitude $A$ hardly varies with swimming speed $U$ during cyclic swimming, whereas $U$ increases linearly with tail-beat frequency $f$. Experimental observations of the correlation between $f$ and $U$ agree with the $\Omega_{min}$ strategy (figure 1). Thus, $U$ should also be proportional to the speed of the curvature wave $w$ ($w = fL$) at $\Omega_{min}$. Generating enough thrust to balance drag during cyclic swimming requires that $w > U$ (figure 1d). This divides the $w$–$U$ space in two zones: a theoretically impossible versus possible zone. Comparison with experiments shows that $U$ is indeed roughly proportional to $w$ across a wide speed- and body-size range. The observed amplitude–frequency combinations cluster rather than occupy the entire feasible region (figure 1d).

The question now arises whether the experimental data cluster close to the predicted $\Omega_{min}$ trajectory. We tested this with the experimental observations and $\Omega_{min}$ computations for larval fishes (figure 4b). First, using all the considered frequency–amplitude combinations, a clear 'empty' zone is present where slip is too low for the thrust-drag balance in the $w$–$U$ plot (figure 1e: insufficient slip zone). The experimental data (white circles) cluster near the black $\Omega_{min}$ curve with energy-optimizing frequency–amplitude combinations, slightly below the 'insufficient slip' boundary. At a given frequency within the parameter space, the fish does not operate at the amplitude that maximizes speed, but a lower amplitude close to the $\Omega_{min}$ trajectory (figure 4b). The fact that fishes use only a small region of the available parameter space near a predicted optimization trajectory suggests that the observed tight correlation between swimming speed and tail-beat frequency results from an optimization strategy rather than fluid-dynamic constraints.

## 4. Discussion

Our results show that fishes minimize speed-specific fluid-dynamic cost of transport ($\Omega_{min}$ strategy) rather than maximize speed-specific Froude efficiency ($\eta_{max}$ strategy). During undulatory swimming, the body wave not only produces thrust but also drag [15], and this drag is substantially higher than in a straight fish moving at the same speed (figure 5b). This high drag explains why undulatory swimmers should favour the $\Omega_{min}$ over the $\eta_{max}$ strategy, substantiating previous studies [8,13]. High tail-beat amplitudes with concomitantly high drag and power requirements may lead to high Froude efficiencies (figures 2b, 3b and 4b), yet do so at the expense of a high hydrodynamic cost of transport. Body shapes with high drag (poorly streamlined) are likewise uneconomical but might still produce high Froude efficiencies.

We propose that minimizing $\Omega$ is the preferred optimization strategy when the energy available to a moving organism or vehicle is decisive. This strategy is most relevant during routine locomotion and can be superseded by other priorities, such as escape efficacy. Deviations from hydrodynamic optimality can also occur owing to physical constraints and trade-offs against other optimality criteria, such as maximizing acceleration, manoeuvrability or overall $\Omega$ (which includes muscle performance). Simulations in laminar flow regime on carangiform and anguilliform swimmers generate similar

results to those of larval fishes, and correctly predict the observed close correlation between tailbeat frequency and swimming speed. Although our simulations are limited to the laminar flow regime, key flow features appear to be robust across flow regimes [27], and the predicted trends occur also in fishes swimming in the turbulent flow regime, suggesting that undulatory swimmers adopt the $\Omega_{\min}$ strategy over a wide range of size and developmental stage. Previous studies argued that fishes swim within a narrow range of Strouhal numbers ($fA/U$) to maximize Froude efficiency [16,32]. Here we show, the observed narrow range of tail-beat amplitudes and Strouhal numbers in cyclic swimming are the result of a $\Omega_{\min}$ strategy.

This study focused on fluid dynamic power, neglecting physiological contributions. Metabolic power is higher because the rate of energy expenditure at rest is not zero (basal metabolic power, $P_{basal}$) and locomotion involves the lossy conversion of chemical into mechanical energy. Yet $P_{swimming}$ (the difference between metabolic power and basal metabolic power) increases with swimming speed [33,34] and therefore positively correlates with mechanical power $P_{mechanical}$. Thus at a specific speed, minimizing $P_{mechanical}$ is equivalent to minimizing $P_{swimming}$, and the speed-specific $\Omega_{\min}$ strategy obtained in this study may minimize speed-specific $\Omega_{metabolic}$. However, owing to physiological contributions, the relationship between metabolic power and speed is unlikely to be monotonic and may in fact be U-shaped [34,35]. In the future, our CFD approach could be combined with models representing the conversion of chemical energy into mechanical work by the swimming musculature.

Our results suggest that the energetic expenditure of undulatory swimming is robust against small deviations in kinematics from the optimum: the energy landscape is approximately flat near the optimum (as evident by the 5% and 10% increment contours in figures 2$b$, 3$b$ and 4$b$). The energy penalty for modest deviations from the $\Omega_{\min}$ trajectory might be low because tail amplitude has a weak effect on speed and $\Omega$ near the $\Omega_{\min}$ trajectory. For example, larval fishes can deviate from the optimal amplitude by 13% (approx. 0.055 $L$) for a mere 5% increase in $\Omega$. This robustness allows larval fishes to use a wide range of amplitudes while they are still developing the neural control of their routine swimming movements. At low swimming speeds, experimental observations usually show a smaller tail-beat amplitude (figure 1$b$ and $c$, data points at the lowest speed side drop sharply) than predicted by CFD. However, this deviation results in a negligible energy penalty because both power output and $\Omega$ are very low at these speeds, and there may exist physiological factors (such as muscle physiology) that prevent fishes from using low-frequency-large-amplitude kinematics.

Our results suggest that, to achieve a speed-specific minimal cost of transport, tail-beat amplitude should be kept within a narrow range, and tail-beat frequency should be changed to control speed. Hence, fish swimming control maybe simpler than previously thought: fishes need not deal with a multi-parameter optimization in the frequency–amplitude parameter space, but can simply focus on adjusting frequency, and can fix tail-beat amplitude near an ideal value. The almost linear relationship between frequency and speed may further simplify speed control. The setpoint of the amplitude depends on the mechanical properties of the fish body and the shape-dependent interactions of the body with the external fluid. The travelling body wave and tail-beat amplitude result from a complex two-way fluid–structure interaction [19,36], which is affected by the flexural stiffness along the body, which in turn depends on passive structural and material properties, and time-dependent contributions of muscle activation [19]. The amplitude setpoint for the $\Omega_{\min}$ strategy depends on the time-dependent bending moment along the fish body, which can be considered as the net actuation along the body. Voesenek *et al.* [19] found that, in larval zebrafish (3–12 dpf), the bending moments appear to be strikingly similar over a wide speed range. Thus, adjusting the tail-amplitude setpoint for $\Omega_{\min}$ may also be relatively simple. The acquired insights on optimization in the swimming kinematics and reduction in the complexity of multi-parameter control in undulatory swimming may inspire research in aquatic organisms and bioinspired robotics using undulatory propulsion.

**Ethics.** The experiment complies to the Dutch Act on Animal Experiments, which complies with European Directive 86/609/EEC, and was approved by the animal welfare authority (DEC Wageningen University, the Netherlands). Institutional licence no. 10 400; protocol no. 2013087.b.

**Data accessibility.** Experimental and computational datasets underpinning the current study, and the code of the three-dimensional Navier-Stokes solver are available in the following Open Science Framework repository: https://doi.org/10.17605/OSF.IO/2VUFG.

**Authors' contributions.** J.L.v.L. provided critical advice on the design of simulations and experiments. G.L. made all simulations. G.L., C.J.V. and J.L.v.L. analysed the CFD results and produced the figures. C.J.V., J.L.v.L. and U.K.M. recorded and analysed the experimental data. H.L. programmed the solution module for the Navier-Stokes equations, whereas G.L. programmed the other CFD modules. U.K.M., G.L. and J.L.v.L. wrote most of the paper and all authors participated in the discussions and revisions.

**Competing interests.** The authors declare no competing interests.

**Funding.** G.L. was funded by the Japan Society for the Promotion of Science (grant nos. JP17K17641 and JP20K14978). H.L. was funded by the Grant-in-Aid for Scientific Research on Innovative Areas of no. 24120007, JSPS. U.K.M. was funded by NSF award 1352130. J.L.v.L. and C.J.V. were funded by NWO-ALW grant no. 824.15.001.

**Acknowledgements.** We would like to thank Otto Berg for providing feedback on the manuscript, Remco Pieters for helping to acquire the experimental data of larval fishes, Eric Tytell for experimental data of eel, and Ramiro Godoy-Diana for experimental data of tetra fish.

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
