## [Peer Review File · Proceedings of the Royal Society B: Biological Sciences]

Review History

RSPB-2021-0618.R0 (Original submission)

Review form: Reviewer 1

Recommendation

Accept with minor revision (please list in comments)

Scientific importance: Is the manuscript an original and important contribution to its field?

Excellent

General interest: Is the paper of sufficient general interest?

Good

Quality of the paper: Is the overall quality of the paper suitable?

Good

Is the length of the paper justified?

Yes

Should the paper be seen by a specialist statistical reviewer?

No

Do you have any concerns about statistical analyses in this paper? If so, please specify them explicitly in your report.

No

It is a condition of publication that authors make their supporting data, code and materials available - either as supplementary material or hosted in an external repository. Please rate, if applicable, the supporting data on the following criteria.

Is it accessible?

Yes

Is it clear?

Yes

Is it adequate?

Yes

Do you have any ethical concerns with this paper?

No

Comments to the Author

Li et al. have performed a study that uses computational fluid dynamics to explore the energetic implications of changing speed through variation in the frequency and amplitude of tail beating in the undulatory swimming of fishes. They found that observed changes in speed among fishes through changes in frequency are consistent with a strategy that minimizes the cost of transport.

I found the design of this study to be innovative and the results are exciting and should interest readers of Proc. B. However, I have concerns about the presentation of this work, mostly through its writing.

The greatest concern I have is how the authors have presented the premise for the study. To my mind, the work successfully tests the hypothesis that speed change through frequency (and constant amplitude) is consistent with minimizing the cost of transport. However, there are passages in the abstract and introduction that give the misleading impression that the authors have assumed, rather than hypothesized, that fishes optimize swimming for efficiency.

Another concern has to do with the definition of the strategy that minimizes Froude efficiency in carangiform swimmers. As shown in Fig. 2b, the trajectory for that strategy lines the border of tail-beat amplitude values in the simulations. I am not clear on what limits that boundary, but regardless, I do not understand how it can be declared as a maximization without any simulations on one side of the trajectory. This issue does not occur in the other swimmers and it is clear that, the live fishes do not operate in that region. Nonetheless, I think there is an issue with the definition of the strategy in the carangiform case.

There are a number of additional areas where I think the writing can be improved.

SPECIFIC COMMENTS

Abstract

It is debatable that animals optimize energetic performance during routine swimming. There are examples both in favor and in opposition to this statement among animals. I recommend leading with a less questionable statement.

“which optimisation strategy” – Not sure what this means. If the authors are assuming optimizing energetic efficiency and the strategy refers to how this is achieved, then there is a

conceptual weakness b/c optimizing for something like efficiency should be a hypothesis and not an assumption. Please at least clarify what is meant by optimization strategy.

What's a "locomotory strategy"? Please define.

"complexity of their kinematic optimisation:" is problematic on multiple fronts. "Complexity" is a tricky term in biology that can mean something that just has yet to be well-explained. Please be clear on the meaning of kinematic optimization. Is something about the kinematics optimized or do the kinematics optimize for some metric of performance. Is this the same thing as optimization strategy?

The science sounds exciting, but the language thus far is confusing and perhaps problematic, if it is assumed, rather than hypothesized, that fish move in an optimal manner.

Introduction

It's a stylistic decision, but embedding equations in the middle of a sentence will unnecessarily alienate some of the broad Proc B readership. For example, one portion could be restated like this: "Analytical models show that thrust is proportional to the square of tail-beat amplitude and frequency, yet fish generally appear to control their speed by vary frequency." It's not really necessary to define terms like density when introducing the reader to the subject of this study.

There are other metrics of "energetic performance", such as the maximum power output. Perhaps it is more accurate to categories COT and Froude efficiency as metrics of efficiency.

"Is the correlation between body-wave frequency and swimming speed caused by energetic optimisation?" – It's a bit odd to stat that energetic optimization causes a pattern of motion when the energetics are in fact caused by that motion. Perhaps rephrase with more biologically sensible causation.

"can be explained by one of two optimisation strategies (maximising η or minimising Ω)" – here I can see a clear meaning for "optimisation strategies"

I like the approach outlined for this study. However, it is not the first to mix and match body shapes, Re, and kinematics (e.g., the work of Petros Koumoutsakos comes to mind: <https://doi.org/10.1017/jfm.2015.283>). An exhausted review is unrealistic for Proc. B, but some recognition of similar approaches in the literature would help place the current work in context.

Methods

Please clarify if novel experiments were run as a component of this study or if the data were previously published. If the work is new, then the experimental methodology should be described in this paper or its supplemental materials. Citing other papers is insufficient.

(e) How was the optimization performed? On Page 8, there is mention of the value of performing an optimization, but it's unclear how the values of frequency and amplitude were arrived at.

Results

It seems to me that Eqn. 1 should be included in the Methods, not Results.

P8 - I am similarly unclear on why the description of optimization methods (and Eqns. 2 and 3) are in Results.

Fig. 1b - Please plot the blue carangiform data in a manner that is visible. That data is currently hidden by the green points. I think it is important, b/c it looks like there may be amplitude

modulation at low speeds.

Fig. 2 - I do not understand how the optimization for optimizing Froude efficiency can occur at the edge of parameter space tested. Isn't it possible that the global optima could be obtained at great values of tail amplitude? If that is not anatomically possible, then it still does not appear accurate to call that a strategy that maximizes the Froude efficiency.

Eqn. 4 and associated text – again, it seems to me that this belongs in the Methods.

“larval fish as example” -> “larval fish as AN example”

(c) Why are thrust and drag calculations being shown? I do not believe there was a justification for this analysis prior to this section of the paper, so please explain what these calculations have to do with the paper's aims.

P11 - Along the same lines, why are the fish being compared with spheres and plates? Please explain the justification for this comparison.

Review form: Reviewer 2

Recommendation

Reject – article is not of sufficient interest (we will consider a transfer to another journal)

Scientific importance: Is the manuscript an original and important contribution to its field?

Good

General interest: Is the paper of sufficient general interest?

Marginal

Quality of the paper: Is the overall quality of the paper suitable?

Good

Is the length of the paper justified?

Yes

Should the paper be seen by a specialist statistical reviewer?

No

Do you have any concerns about statistical analyses in this paper? If so, please specify them explicitly in your report.

No

It is a condition of publication that authors make their supporting data, code and materials available - either as supplementary material or hosted in an external repository. Please rate, if applicable, the supporting data on the following criteria.

Is it accessible?

Yes

Is it clear?

No

Is it adequate?

Yes

Do you have any ethical concerns with this paper?

No

Comments to the Author

This manuscript, "Fish regulate tail-beat kinematics to minimize speed-specific cost of transport," examined the relationship between tail kinematics, frequency and amplitude, with cost of transport and Froude efficiency. The authors found that based on simulations and results from experimental fish, that fish use a strategy of tail beat frequency and amplitude to minimize cost of transport, not Froude efficiency. This is a well-written manuscript with interesting results. I appreciate the comparison of cost of transport with Froude efficiency during undulatory locomotion. The addition of some data from experimental animals is a benefit. However, the scope of the paper is a bit narrow and lacks comparisons to physiological studies examining the cost of transport and studies of smaller swimming fish.

My main suggestion for the manuscript would be to incorporate more examples from living fish for the cost of transport. It would be interesting to know how the values obtained in this study match those of living fish. Cost of transport can be determined several ways from studies examining the metabolic rate of an animal at increasing speeds and it is not clear where the energy consumption in the equation for cost of transport in the supplementary information came from. Since the paper focuses on comparing cost of transport to Froude efficiency, more detail is needed on how cost of transport was obtained in the current study and how it relates to studies of living fish.

In terms of tail kinematics, Figure 1 is a nice example of the relationship of tail beat frequency in relation to size of fish and swimming speed, but it would be nice to see this relationship for tail beat amplitude. Since the authors suggest fish should fix tail beat amplitude and adjust frequency to change speed, a figure of amplitude against speed for the species in Figure 1a would strengthen that result. Furthermore, it is mentioned in the manuscript that in the model tail beat amplitude is affected by tail beat frequency, but Figure 1b shows there is no relationship between the two. It is not clear if there should be a relationship and how this dependency in the model affects the results.

My last main comment is to have more justification why a tetra was used as a model for a carangiform swimmer? Many of the species in Figure 1a are larger and the larvae was modeled after a zebrafish, so it wasn't clear why a smaller tetra species was used. Adding examples of smaller species to Figure 1a would be helpful for comparisons, or modeling a carangiform swimmer after a larger species would be an interesting comparison to see how the model holds up.

Some minor comments:

In the methods, under (e): while terminal speed is defined, it is still a bit unclear. It is mentioned in the discussion that at low speeds, fish deviate further from the optimal amplitude than at high speeds, can you provide a reference?

Decision letter (RSPB-2021-0618.R0)

09-Apr-2021

Dear Dr Li:

I am writing to inform you that your manuscript RSPB-2021-0618 entitled "Fish regulate tail-beat kinematics to minimise speed-specific cost of transport" has, in its current form, been rejected for publication in Proceedings B.

This action has been taken on the advice of referees, who have recommended that substantial revisions are necessary. With this in mind we would be happy to consider a resubmission, provided the comments of the referees are fully addressed. However please note that this is not a provisional acceptance.

In review, it was raised that it was not clear how cost of transport was modelled. Furthermore, it is not clear just how reproducible the CFD analyses are-- is the community expected to reconstruct them from multiple papers, whereas they would be better served, and our open science policy met, by supplying the final files directly? Please address this carefully in a resubmission.

Sincerely,

Dr John Hutchinson, Editor
mailto: proceedingsb@royalsociety.org

Associate Editor

Comments to Author:

Associate Editor: Doug Altshuler

This manuscript takes a broad perspective on analyzing fish swim kinematics. It includes an novel application of CFD to the problem, and the figures paint an interesting picture. I was not especially convinced by the expressed motivation for the work, and struggled with the presentation in multiple sections. The two referees brought up a number of substantive concerns although they were not always in alignment in regards to the strengths and weaknesses. It seems possible that the range of concerns could be addressed through a major revision.

Reviewer(s)' Comments to Author:

Referee: 1

Comments to the Author(s)

Li et al. have performed a study that uses computational fluid dynamics to explore the energetic implications of changing speed through variation in the frequency and amplitude of tail beating in the undulatory swimming of fishes. They found that observed changes in speed among fishes through changes in frequency are consistent with a strategy that minimizes the cost of transport.

I found the design of this study to be innovative and the results are exciting and should interest readers of Proc. B. However, I have concerns about the presentation of this work, mostly through its writing.

The greatest concern I have is how the authors have presented the premise for the study. To my mind, the work successfully tests the hypothesis that speed change through frequency (and constant amplitude) is consistent with minimizing the cost of transport. However, there are passages in the abstract and introduction that give the misleading impression that the authors have assumed, rather than hypothesized, that fishes optimize swimming for efficiency.

Another concern has to do with the definition of the strategy that minimizes Froude efficiency in carangiform swimmers. As shown in Fig. 2b, the trajectory for that strategy lines the border of tail-beat amplitude values in the simulations. I am not clear on what limits that boundary, but regardless, I do not understand how it can be declared as a maximization without any simulations on one side of the trajectory. This issue does not occur in the other swimmers and it is clear that, the live fishes do not operate in that region. Nonetheless, I think there is an issue with the definition of the strategy in the carangiform case.

There are a number of additional areas where I think the writing can be improved.

SPECIFIC COMMENTS

Abstract

It is debatable that animals optimize energetic performance during routine swimming. There are examples both in favor and in opposition to this statement among animals. I recommend leading with a less questionable statement.

“which optimisation strategy” – Not sure what this means. If the authors are assuming optimizing energetic efficiency and the strategy refers to how this is achieved, then there is a conceptual weakness b/c optimizing for something like efficiency should be a hypothesis and not an assumption. Please at least clarify what is meant by optimization strategy.

What's a “locomotory strategy”? Please define.

“complexity of their kinematic optimisation:” is problematic on multiple fronts. “Complexity” is a tricky term in biology that can mean something that just has yet to be well-explained. Please be clear on the meaning of kinematic optimization. Is something about the kinematics optimized or do the kinematics optimize for some metric of performance. Is this the same thing as optimization strategy?

The science sounds exciting, but the language thus far is confusing and perhaps problematic, if it is assumed, rather than hypothesized, that fish move in an optimal manner.

Introduction

It's a stylistic decision, but embedding equations in the middle of a sentence will unnecessarily alienate some of the broad Proc B readership. For example, one portion could be restated like

this: “Analytical models show that thrust is proportional to the square of tail-beat amplitude and frequency, yet fish generally appear to control their speed by vary frequency.” It’s not really necessary to define terms like density when introducing the reader to the subject of this study.

There are other metrics of “energetic performance”, such as the maximum power output. Perhaps it is more accurate to categories COT and Froude efficiency as metrics of efficiency.

“Is the correlation between body-wave frequency and swimming speed caused by energetic optimisation?” – It’s a bit odd to stat that energetic optimization causes a pattern of motion when the energetics are in fact caused by that motion. Perhaps rephrase with more biologically sensible causation.

“can be explained by one of two optimisation strategies (maximising η or minimising Ω)” – here I can see a clear meaning for “optimisation strategies”

I like the approach outlined for this study. However, it is not the first to mix and match body shapes, Re, and kinematics (e.g., the work of Petros Koumoutsakos comes to mind: <https://doi.org/10.1017/jfm.2015.283>). An exhausted review is unrealistic for Proc. B, but some recognition of similar approaches in the literature would help place the current work in context.

Methods

Please clarify if novel experiments were run as a component of this study or if the data were previously published. If the work is new, then the experimental methodology should be described in this paper or its supplemental materials. Citing other papers is insufficient.

(e) How was the optimization performed? On Page 8, there is mention of the value of performing an optimization, but it’s unclear how the values of frequency and amplitude were arrived at.

Results

It seems to me that Eqn. 1 should be included in the Methods, not Results.

P8 - I am similarly unclear on why the description of optimization methods (and Eqns. 2 and 3) are in Results.

Fig. 1b - Please plot the blue carangiform data in a manner that is visible. That data is currently hidden by the green points. I think it is important, b/c it looks like there may be amplitude modulation at low speeds.

Fig. 2 - I do not understand how the optimization for optimizing Froude efficiency can occur at the edge of parameter space tested. Isn’t it possible that the global optima could be obtain at great values of tail amplitude? If that is not anatomically possible, then it still does not appear accurate to call that a strategy that maximizes the Froude efficiency.

Eqn. 4 and associated text – again, it seems to me that this belongs in thee Methods.

“larval fish as example” -> “larval fish as AN example”

(c) Why are thrust and drag calculations being shown? I do not believe there was a justification for this analysis prior to this section of the paper, so please explain what these calculations have to do with the paper’s aims.

P11 - Along the same lines, why are the fish being compared with spheres and plates? Please explain the justification for this comparison.

Referee: 2

Comments to the Author(s)

This manuscript, "Fish regulate tail-beat kinematics to minimize speed-specific cost of transport," examined the relationship between tail kinematics, frequency and amplitude, with cost of transport and Froude efficiency. The authors found that based on simulations and results from experimental fish, that fish use a strategy of tail beat frequency and amplitude to minimize cost of transport, not Froude efficiency. This is a well-written manuscript with interesting results. I appreciate the comparison of cost of transport with Froude efficiency during undulatory locomotion. The addition of some data from experimental animals is a benefit. However, the scope of the paper is a bit narrow and lacks comparisons to physiological studies examining the cost of transport and studies of smaller swimming fish.

My main suggestion for the manuscript would be to incorporate more examples from living fish for the cost of transport. It would be interesting to know how the values obtained in this study match those of living fish. Cost of transport can be determined several ways from studies examining the metabolic rate of an animal at increasing speeds and it is not clear where the energy consumption in the equation for cost of transport in the supplementary information came from. Since the paper focuses on comparing cost of transport to Froude efficiency, more detail is needed on how cost of transport was obtained in the current study and how it relates to studies of living fish.

In terms of tail kinematics, Figure 1 is a nice example of the relationship of tail beat frequency in relation to size of fish and swimming speed, but it would be nice to see this relationship for tail beat amplitude. Since the authors suggest fish should fix tail beat amplitude and adjust frequency to change speed, a figure of amplitude against speed for the species in Figure 1a would strengthen that result. Furthermore, it is mentioned in the manuscript that in the model tail beat amplitude is affected by tail beat frequency, but Figure 1b shows there is no relationship between the two. It is not clear if there should be a relationship and how this dependency in the model affects the results.

My last main comment is to have more justification why a tetra was used as a model for a carangiform swimmer? Many of the species in Figure 1a are larger and the larvae was modeled after a zebrafish, so it wasn't clear why a smaller tetra species was used. Adding examples of smaller species to Figure 1a would be helpful for comparisons, or modeling a carangiform swimmer after a larger species would be an interesting comparison to see how the model holds up.

Some minor comments:

In the methods, under (e): while terminal speed is defined, it is still a bit unclear.

It is mentioned in the discussion that at low speeds, fish deviate further from the optimal amplitude than at high speeds, can you provide a reference?

Author's Response to Decision Letter for (RSPB-2021-0618.R0)

See Appendix A.

RSPB-2021-1601.R0

Review form: Reviewer 1

Recommendation

Accept as is

Scientific importance: Is the manuscript an original and important contribution to its field?

Good

General interest: Is the paper of sufficient general interest?

Good

Quality of the paper: Is the overall quality of the paper suitable?

Excellent

Is the length of the paper justified?

Yes

Should the paper be seen by a specialist statistical reviewer?

No

Do you have any concerns about statistical analyses in this paper? If so, please specify them explicitly in your report.

No

It is a condition of publication that authors make their supporting data, code and materials available - either as supplementary material or hosted in an external repository. Please rate, if applicable, the supporting data on the following criteria.

Is it accessible?

Yes

Is it clear?

Yes

Is it adequate?

Yes

Do you have any ethical concerns with this paper?

Yes

Comments to the Author

I am satisfied with the authors' responses to my concerns and those of the other reviewer.

Review form: Reviewer 3

Recommendation

Accept with minor revision (please list in comments)

Scientific importance: Is the manuscript an original and important contribution to its field?

Excellent

General interest: Is the paper of sufficient general interest?

Excellent

Quality of the paper: Is the overall quality of the paper suitable?

Excellent

Is the length of the paper justified?

Yes

Should the paper be seen by a specialist statistical reviewer?

No

Do you have any concerns about statistical analyses in this paper? If so, please specify them explicitly in your report.

No

It is a condition of publication that authors make their supporting data, code and materials available - either as supplementary material or hosted in an external repository. Please rate, if applicable, the supporting data on the following criteria.

Is it accessible?

Yes

Is it clear?

Yes

Is it adequate?

Yes

Do you have any ethical concerns with this paper?

No

Comments to the Author

General comments:

This manuscript looks at possible optimization criteria to explain the kinematics used by fishes during steady swimming. The simulations are fairly comprehensive, the question is of broad interest, and I would like to see this paper published. I have a few minor concerns.

- 1) Generally, I question scaling by body length, especially when body shapes vary. The authors will probably do this anyway, and as it is consistent with the vast majority of the literature, I am willing to let it slide.
- 2) The phrasing of the main questions in the introduction seem to assume a priori some kind of optimization strategy (the two being studied as contenders), when in reality, it is theoretically possible for the fish not to be optimizing anything.
- 3) I think the authors should weaken their claim that these findings apply to all fishes. Given the Reynolds regime studied, and the lack of extensive data backing up the assumption that laminar flow regimes are generally similar to turbulent flow regimes in fishes, I think some discussion of this limitation is appropriate.

Specific comments:

Ln 49: I understand the desire to make things unitless and normalize for body size, but I generally question the strategy of doing this by dividing by body length. Yes, there is precedent for it in the literature, but when you are dealing with fishes of radically different shapes/aspect

ratios, does it make sense to do this? Is dividing by body length for a 2cm eel the same as dividing by body length for a 2cm tetra?

Ln 64-67: It seems to me that, if you are not assuming energy optimization outright (which I hope the authors are not), question 2 is 100% predicated on the answer to question 1 being “yes”. The way question 2 is phrased also sounds like fishes MUST be optimizing in one of those two ways... even though, hypothetically, both could be false. I would suggest reframing these questions to avoid upfront assumptions about optimization, and allow for the possibility of other explanations. For question 2, the word “if” would go a long way.

Ln 74: It is a bit strong to suggest that the data in this study apply to all fish when the size of fishes tested was only 2cm, when that is still in a laminar regime – though I understand the complications involved with simulating turbulent flows.

Ln 75: Sounds like there is a correlation/causation issue here. The data in this study support that fishes are minimizing mechanical cost of transport, but they do not exclude the possibility that fishes are optimizing something else (metabolic CoT). This is mentioned in the discussion, but perhaps could be mentioned earlier.

Ln 83 and throughout: The supplementary materials are extensive (and very clear. Thanks!!!). It would be incredibly useful to the reader if the authors indicate which section of the supplement they are referring to at a given place in the manuscript.

Ln 85: I am not qualified to comment on the CFD modeling methods.

Ln 95: Do tetra swim using carangiform kinematics? At least in aquaria, they seem to be intermittent swimmers. It would be useful to see a reference for this, if one exists.

Ln 98-100: Were these kinematics, or the resulting kinematics from the simulations, compared to those of actual fishes swimming? Would be useful to know how near a match they are.

Ln 121-122: Again, I am wondering how robust these findings are to changes in body size. The reference provided here, Liu et al 2017, does not itself provide evidence that this is the case, though it contains a discussion of literature. One of the studies cited as evidence for the similarity of laminar vs turbulent flow fields (Müller et al. 2001) does not appear to directly address this – if it does provide evidence of this similarity, it is not clear. Furthermore, Müller 2001 only generated 2D flow visualizations, making it unsuitable for comparison to a 3D simulation. The reference Müller 2001 is being compared to (Kern & Koumoutsakos) is only a simulation for the anguilliform case. All of this is to say that, while I understand using low Re simulations for computational feasibility, in the absence of more data, I am less certain of their application to larger fish. This limitation should at least be mentioned and discussed, rather than dismissed out of hand. Unless I am grossly misinterpreting these studies.

Ln 178-189: Looking back at the figures showing the performance surfaces for CoT and Froude efficiency, it looks like the combinations of frequency and amplitude that are good for CoT minimizing are explicitly bad for Froude maximization and vice versa. Is there a reason to apriori expect there to be a tradeoff between the two? If yes, it would be great to see that in the introduction and/or the discussion.

Ln 200-205: It would be useful to see a depiction of different values of beta.

Ln 231-232: How were these “physiologically realistic limits” determined?

Ln 238: I find the use of equations in parentheses in sentences, especially when two equations for opposite things are in series like this, to be quite difficult to parse.

Ln 245-250: The experimental data look like there is a curvilinear (maybe quadratic?) relationship between w and U , as opposed to a linear one. Is there an explanation for that? I think what I am saying is I disagree with the statement that U is roughly proportional with w .

Ln 256-258: I find a lot of things about this sentence's phrasing troubling. I highly doubt that fish operate at a "given" frequency, and they certainly are not "choosing" an amplitude to "maximize speed". If anything, the fish "chooses" a speed, and modulates frequency. The order of operations here does not make sense. I am not sure I understand how these conclusions are being drawn, but that may be a limitation of my own knowledge. That said, for a Proc B readership, perhaps this should be explained more clearly.

Ln 272-292: I love these paragraphs. But, in line 287, I wonder about the use of these references to support the claim that $P_{\text{mechanical}}$ and $P_{\text{metabolic}}$ are highly correlated. It is a very reasonable assumption – but neither of these studies measured mechanical power output. So, I would say that it is a safe assumption, but not known.

Ln 303-304: Can the authors (very briefly) explain how muscle fiber types would preclude low-frequency large-amplitude kinematics? I can come up with an explanation, but the reader may not.

Figure 1a: I have a very hard time distinguishing the colors of the eel, trout and dace, and the jack mackerel, carp and dace points. Can the authors maybe use different shapes in addition to color?

Figures 2-4: This may just be me being picky, but in general I consider brighter colors to be "better". For the c,d,e panels, this results in a different frame of reference: while yellow agrees with my expectation in c and e, d is the opposite of what I expect. I spent a lot of time completely misinterpreting the d panels because of the color scheme. Maybe the authors could flip the color palette for panel d? But this may not be a concern for other people.

Supplemental materials: well organized, well presented and very helpful. I do wish some of these were included in the main text, but I understand space limitations here.

Review form: Reviewer 4

Recommendation

Major revision is needed (please make suggestions in comments)

Scientific importance: Is the manuscript an original and important contribution to its field?

Good

General interest: Is the paper of sufficient general interest?

Good

Quality of the paper: Is the overall quality of the paper suitable?

Acceptable

Is the length of the paper justified?

Yes

Should the paper be seen by a specialist statistical reviewer?

No

Do you have any concerns about statistical analyses in this paper? If so, please specify them explicitly in your report.

No

It is a condition of publication that authors make their supporting data, code and materials available - either as supplementary material or hosted in an external repository. Please rate, if applicable, the supporting data on the following criteria.

Is it accessible?

Yes

Is it clear?

Yes

Is it adequate?

Yes

Do you have any ethical concerns with this paper?

Yes

Comments to the Author

This study used numerical methods to ask whether fish modulate their swimming kinematics (tail beat amplitude and frequency) to minimize their energetic expenditure, and specifically do they minimize the cost of transport or maximize Froude efficiency. The authors generate numerical simulations of swimming fishes and examine how the cost of transport and Froude efficiency change through the parameter range (tail beat amplitude and frequency). They further infer the areas in which these metrics are optimized and ask whether these areas fit the observed range of swimming kinematics in fishes.

The question presented is central and timely in fish biology. More broadly, whether and how animals optimize their morphology and behavior is of broad interest. I am a fan of the approach presented in this paper, which combines numerical simulations and observations. Therefore, this study can be of interest to the broad readership of ProcB, but some of the parts will have to be re-written to be suitable for this readership. Specifically, the intro is inaccessible to non-experts (as is immediately apparent with the use technical terms such as “dimensionless tail-beat amplitude” in the first paragraph). The introduction must provide the general reader some sort of intuition about the problem, and gradually introduce them to the technical terms such as “cycle averages of respectively thrust, swimming speed, and input power”. Also, the paper dwells quickly into the question of which metric of optimization is more important, but fails to explain why this is a general and interesting question in biology (and it is!). Another issue with the introduction is that it glosses over previous attempts of dealing with this question, some of which were carried out by authors of this study but also by others. there should be an explanation of why this study is needed in the light of these previous attempts. I understand that there is a word limitation in this journal, but I see these issues as essential.

A second issue is related to the limits of interpretation. In general, the parameter ranges for the modeled and observed data only partially overlap. This limitation is not discussed, and comparisons are made irrespective of it. This seems like a major problem in the design of the study and it should be amended. Either expand the performance maps to match the range of observed data, or collect data that would fit the current maps. Below I expand on this point:

For all the right reasons, numerical simulations were carried out for a range of Reynolds numbers of <2000 , limiting body size to 20 mm. However, except for larval zebrafish, none of the fish for which data is provided swims at such low Re/small size. Thus, most of the data presented as evidence for the fit between the numerical solution and the observations is outside the parameter range of the simulation and is therefore a gross extrapolation. Fig 1 clearly shows that size is a

major determinant of frequency range, but this is ignored in the interpretation of the results. Given that collecting data on the frequency and amplitude of swimming fishes is not a highly demanding task, the immediate solution to this would be to obtain experimental data that fits the parameter range of the simulations. Even after this essential addition, the authors must acknowledge the limitations of their data, and avoid over-interpretation to domains outside of their parameter range. That is, applying the results to “all fish” warrants at least some caution.

Along the same lines, what determined the range of relative speeds modeled for the different swimmers (Fig 2-4)? Was there any attempt to match the range to the parameter range observed in nature? A superficial examination of Fig 1C reveals that the modeled range for anguilliform exceeds that presented in the data by 2-fold while that for carangiforms is narrower by 50%. Specifically, $\frac{3}{4}$ of the points for Tetra fall outside of the modeled line. How is this justified? Surely this will affect interpretation/confidence?

There are several additional issues that need to be addressed in revision this paper, mostly related to the relevance of the parameter range of the computational work to the real-world data:

- 1) The range of relative speeds in the carangiform and anguilliform swimmers compared to the larval fish is perplexing. Surely larval zebrafish are not the fastest fish in the world (in terms of BL/s)? if the speed of the other fish is capped due to computational limitations, than this again dampens the ability to really understand which metric is optimized at relative speeds that are relevant to the real world.
- 2) Can larval zebrafish be classified on the anguilliform-carangiform spectrum? Also, the range of speeds given here is ~2-5 fold greater than reported in other studies of zebrafish swimmers. can you explain this?
- 3) Fig 1 is beautiful but can be confusing. It is unclear what are the gray lines are, and why some genus appear twice (e.g. trout) while others appear once. Panels a and b are redundant- pane B should be in the ESM. Panel e (and also d to a lesser extent) does not really belong to the panel.
- 4) Panels c2 and c3 give the impression that speed and frequency are correlated (BTW, why are the axes flipped compared to panel a?). What would be the results of a multiple regression model that took into account the effects of amplitude and frequency (and perhaps size) on speed? I am guessing all three will be significant.
- 5) Denoting the ± 5 and 10% lines for the minimum-cost-of-transport curve in Fig 2-4 is essential, but why is are the same lines omitted for Froude efficiency?
- 6) The explanation of how the curves for minimum-cost-of-transport and Froude efficiency are drawn should be in the paper and not in the (excellent and detailed) ESM file.
- 7) Also, why does the calculation of the two curves takes into account speeds? I am not sure that I understand this from the explanation in lines 55-60. both metrics have U in them already?
- 8) why is amplitude normalized but frequency isn't?

Decision letter (RSPB-2021-1601.R0)

07-Sep-2021

Dear Dr Li:

Your manuscript has now been peer reviewed and the reviews have been assessed by an Associate Editor. The reviewers' comments (not including confidential comments to the Editor) and the comments from the Associate Editor are included at the end of this email for your reference. As you will see, the reviewers and the Editors have raised some concerns with your manuscript and we would like to invite you to revise your manuscript to address them.

Research ethics:

Use of animals and field studies:

It is a condition of publication that you make available the data and research materials supporting the results in the article (<https://royalsociety.org/journals/authors/author-guidelines/#data>). Datasets should be deposited in an appropriate publicly available repository and details of the associated accession number, link or DOI to the datasets must be included in the Data Accessibility section of the article (<https://royalsociety.org/journals/ethics-policies/data-sharing-mining/>). Reference(s) to datasets should also be included in the reference list of the article with DOIs (where available).

Please submit a copy of your revised paper within three weeks. If we do not hear from you within this time your manuscript will be rejected. If you are unable to meet this deadline please let us know as soon as possible, as we may be able to grant a short extension.

Best wishes,
Dr John Hutchinson, Editor
mailto:proceedingsb@royalsociety.org

Associate Editor Board Member
Comments to Author:
Associate Editor: Doug Altshuler

Li et al. have resubmitted a manuscript that is responsive to the previous reviews. This is a very interesting study that has broad explanatory power to understand the role of swimming kinematics in energy expenditure. I regret that it took extra time for this next round of review. One of the previous referees was unavailable. It took some effort to secure a second referee and in the end we would up with two of them. All of the assessments were excellent and they contain a number of suggestions that should help make the manuscript more accessible to a general audience. I believe it would be valuable to incorporate many of the suggested changes. I do not believe that any new data are required or that new analysis is needed. Some adjustments to the text, especially the introduction, and perhaps some adjustments to the figures as referee #3 suggests will be particularly helpful.

Reviewer(s)' Comments to Author:

Referee: 1

Comments to the Author(s).

I am satisfied with the authors' responses to my concerns and those of the other reviewer.

Referee: 3

Comments to the Author(s).

General comments:

This manuscript looks at possible optimization criteria to explain the kinematics used by fishes during steady swimming. The simulations are fairly comprehensive, the question is of broad interest, and I would like to see this paper published. I have a few minor concerns.

- 1) Generally, I question scaling by body length, especially when body shapes vary. The authors will probably do this anyway, and as it is consistent with the vast majority of the literature, I am willing to let it slide.
- 2) The phrasing of the main questions in the introduction seem to assume a priori some kind of optimization strategy (the two being studied as contenders), when in reality, it is theoretically possible for the fish not to be optimizing anything.
- 3) I think the authors should weaken their claim that these findings apply to all fishes. Given the Reynolds regime studied, and the lack of extensive data backing up the assumption that laminar flow regimes are generally similar to turbulent flow regimes in fishes, I think some discussion of this limitation is appropriate.

Specific comments:

Ln 49: I understand the desire to make things unitless and normalize for body size, but I generally question the strategy of doing this by dividing by body length. Yes, there is precedent for it in the literature, but when you are dealing with fishes of radically different shapes/aspect ratios, does it make sense to do this? Is dividing by body length for a 2cm eel the same as dividing by body length for a 2cm tetra?

Ln 64-67: It seems to me that, if you are not assuming energy optimization outright (which I hope the authors are not), question 2 is 100% predicated on the answer to question 1 being “yes”. The way question 2 is phrased also sounds like fishes MUST be optimizing in one of those two ways... even though, hypothetically, both could be false. I would suggest reframing these questions to avoid upfront assumptions about optimization, and allow for the possibility of other explanations. For question 2, the word “if” would go a long way.

Ln 74: It is a bit strong to suggest that the data in this study apply to all fish when the size of fishes tested was only 2cm, when that is still in a laminar regime – though I understand the complications involved with simulating turbulent flows.

Ln 75: Sounds like there is a correlation/causation issue here. The data in this study support that fishes are minimizing mechanical cost of transport, but they do not exclude the possibility that fishes are optimizing something else (metabolic CoT). This is mentioned in the discussion, but perhaps could be mentioned earlier.

Ln 83 and throughout: The supplementary materials are extensive (and very clear. Thanks!!!). It would be incredibly useful to the reader if the authors indicate which section of the supplement they are referring to at a given place in the manuscript.

Ln 85: I am not qualified to comment on the CFD modeling methods.

Ln 95: Do tetra swim using carangiform kinematics? At least in aquaria, they seem to be intermittent swimmers. It would be useful to see a reference for this, if one exists.

Ln 98-100: Were these kinematics, or the resulting kinematics from the simulations, compared to those of actual fishes swimming? Would be useful to know how near a match they are.

Ln 121-122: Again, I am wondering how robust these findings are to changes in body size. The reference provided here, Liu et al 2017, does not itself provide evidence that this is the case, though it contains a discussion of literature. One of the studies cited as evidence for the similarity of laminar vs turbulent flow fields (Müller et al. 2001) does not appear to directly address this – if it does provide evidence of this similarity, it is not clear. Furthermore, Müller 2001 only generated 2D flow visualizations, making it unsuitable for comparison to a 3D simulation. The reference Müller 2001 is being compared to (Kern & Koumoutsakos) is only a simulation for the anguilliform case. All of this is to say that, while I understand using low Re simulations for computational feasibility, in the absence of more data, I am less certain of their application to

larger fish. This limitation should at least be mentioned and discussed, rather than dismissed out of hand. Unless I am grossly misinterpreting these studies.

Ln 178-189: Looking back at the figures showing the performance surfaces for CoT and Froude efficiency, it looks like the combinations of frequency and amplitude that are good for CoT minimizing are explicitly bad for Froude maximization and vice versa. Is there a reason to a priori expect there to be a tradeoff between the two? If yes, it would be great to see that in the introduction and/or the discussion.

Ln 200-205: It would be useful to see a depiction of different values of beta.

Ln 231-232: How were these “physiologically realistic limits” determined?

Ln 238: I find the use of equations in parentheses in sentences, especially when two equations for opposite things are in series like this, to be quite difficult to parse.

Ln 245-250: The experimental data look like there is a curvilinear (maybe quadratic?) relationship between w and U , as opposed to a linear one. Is there an explanation for that? I think what I am saying is I disagree with the statement that U is roughly proportional with w .

Ln 256-258: I find a lot of things about this sentence’s phrasing troubling. I highly doubt that fish operate at a “given” frequency, and they certainly are not “choosing” an amplitude to “maximize speed”. If anything, the fish “chooses” a speed, and modulates frequency. The order of operations here does not make sense. I am not sure I understand how these conclusions are being drawn, but that may be a limitation of my own knowledge. That said, for a Proc B readership, perhaps this should be explained more clearly.

Ln 272-292: I love these paragraphs. But, in line 287, I wonder about the use of these references to support the claim that $P_{\text{mechanical}}$ and $P_{\text{metabolic}}$ are highly correlated. It is a very reasonable assumption – but neither of these studies measured mechanical power output. So, I would say that it is a safe assumption, but not known.

Ln 303-304: Can the authors (very briefly) explain how muscle fiber types would preclude low-frequency large-amplitude kinematics? I can come up with an explanation, but the reader may not.

Figure 1a: I have a very hard time distinguishing the colors of the eel, trout and dace, and the jack mackerel, carp and dace points. Can the authors maybe use different shapes in addition to color?

Figures 2-4: This may just be me being picky, but in general I consider brighter colors to be “better”. For the c,d,e panels, this results in a different frame of reference: while yellow agrees with my expectation in c and e, d is the opposite of what I expect. I spent a lot of time completely misinterpreting the d panels because of the color scheme. Maybe the authors could flip the color palette for panel d? But this may not be a concern for other people.

Supplemental materials: well organized, well presented and very helpful. I do wish some of these were included in the main text, but I understand space limitations here.

Referee: 4

Comments to the Author(s).

This study used numerical methods to ask whether fish modulate their swimming kinematics (tail beat amplitude and frequency) to minimize their energetic expenditure, and specifically do they minimize the cost of transport or maximize Froude efficiency. The authors generate numerical simulations of swimming fishes and examine how the cost of transport and Froude efficiency change through the parameter range (tail beat amplitude and frequency). They further

infer the areas in which these metrics are optimized and ask whether these areas fit the observed range of swimming kinematics in fishes.

The question presented is central and timely in fish biology. More broadly, whether and how animals optimize their morphology and behavior is of broad interest. I am a fan of the approach presented in this paper, which combines numerical simulations and observations. Therefore, this study can be of interest to the broad readership of *ProcB*, but some of the parts will have to be rewritten to be suitable for this readership. Specifically, the intro is inaccessible to non-experts (as is immediately apparent with the use of technical terms such as “dimensionless tail-beat amplitude” in the first paragraph). The introduction must provide the general reader some sort of intuition about the problem, and gradually introduce them to the technical terms such as “cycle averages of respectively thrust, swimming speed, and input power”. Also, the paper dwells quickly into the question of which metric of optimization is more important, but fails to explain why this is a general and interesting question in biology (and it is!). Another issue with the introduction is that it glosses over previous attempts of dealing with this question, some of which were carried out by authors of this study but also by others. There should be an explanation of why this study is needed in the light of these previous attempts. I understand that there is a word limitation in this journal, but I see these issues as essential.

A second issue is related to the limits of interpretation. In general, the parameter ranges for the modeled and observed data only partially overlap. This limitation is not discussed, and comparisons are made irrespective of it. This seems like a major problem in the design of the study and it should be amended. Either expand the performance maps to match the range of observed data, or collect data that would fit the current maps. Below I expand on this point:

For all the right reasons, numerical simulations were carried out for a range of Reynolds numbers of <2000 , limiting body size to 20 mm. However, except for larval zebrafish, none of the fish for which data is provided swims at such low Re /small size. Thus, most of the data presented as evidence for the fit between the numerical solution and the observations is outside the parameter range of the simulation and is therefore a gross extrapolation. Fig 1 clearly shows that size is a major determinant of frequency range, but this is ignored in the interpretation of the results. Given that collecting data on the frequency and amplitude of swimming fishes is not a highly demanding task, the immediate solution to this would be to obtain experimental data that fits the parameter range of the simulations. Even after this essential addition, the authors must acknowledge the limitations of their data, and avoid over-interpretation to domains outside of their parameter range. That is, applying the results to “all fish” warrants at least some caution.

Along the same lines, what determined the range of relative speeds modeled for the different swimmers (Fig 2-4)? Was there any attempt to match the range to the parameter range observed in nature? A superficial examination of Fig 1C reveals that the modeled range for anguilliform exceeds that presented in the data by 2-fold while that for carangiforms is narrower by 50%. Specifically, $\frac{3}{4}$ of the points for Tetra fall outside of the modeled line. How is this justified? Surely this will affect interpretation/confidence?

There are several additional issues that need to be addressed in revision this paper, mostly related to the relevance of the parameter range of the computational work to the real-world data:

- 1) The range of relative speeds in the carangiform and anguilliform swimmers compared to the larval fish is perplexing. Surely larval zebrafish are not the fastest fish in the world (in terms of BL/s)? If the speed of the other fish is capped due to computational limitations, then this again dampens the ability to really understand which metric is optimized at relative speeds that are relevant to the real world.
- 2) Can larval zebrafish be classified on the anguilliform-carangiform spectrum? Also, the range of speeds given here is ~ 2 -5 fold greater than reported in other studies of zebrafish swimmers. Can you explain this?

- 3) Fig 1 is beautiful but can be confusing. It is unclear what are the gray lines are, and why some genus appear twice (e.g. trout) while others appear once. Panels a and b are redundant- pane B should be in the ESM. Panel e (and also d to a lesser extent) does not really belong to the panel.
- 4) Panels c2 and c3 give the impression that speed and frequency are correlated (BTW, why are the axes flipped compared to panel a?). What would be the results of a multiple regression model that took into account the effects of amplitude and frequency (and perhaps size) on speed? I am guessing all three will be significant.
- 5) Denoting the ± 5 and 10% lines for the minimum-cost-of-transport curve in Fig 2-4 is essential, but why are the same lines omitted for Froude efficiency?
- 6) The explanation of how the curves for minimum-cost-of-transport and Froude efficiency are drawn should be in the paper and not in the (excellent and detailed) ESM file.
- 7) Also, why does the calculation of the two curves takes into account speeds? I am not sure that I understand this from the explanation in lines 55-60. both metrics have U in them already?
- 8) why is amplitude normalized but frequency isn't?

Author's Response to Decision Letter for (RSPB-2021-1601.R0)

See Appendix B.

RSPB-2021-1601.R1

Review form: Reviewer 1

Recommendation

Accept as is

Scientific importance: Is the manuscript an original and important contribution to its field?

Excellent

General interest: Is the paper of sufficient general interest?

Excellent

Quality of the paper: Is the overall quality of the paper suitable?

Excellent

Is the length of the paper justified?

Yes

Should the paper be seen by a specialist statistical reviewer?

No

Do you have any concerns about statistical analyses in this paper? If so, please specify them explicitly in your report.

No

It is a condition of publication that authors make their supporting data, code and materials available - either as supplementary material or hosted in an external repository. Please rate, if applicable, the supporting data on the following criteria.

Is it accessible?

Yes

Is it clear?

Yes

Is it adequate?

Yes

Do you have any ethical concerns with this paper?

No

Comments to the Author

This looks good. I am satisfied with the changes made, and the reasons given where changes were not made.

Decision letter (RSPB-2021-1601.R1)

04-Nov-2021

Dear Dr Li

I am pleased to inform you that your manuscript entitled "Fish regulate tail-beat kinematics to minimise speed-specific cost of transport" has been accepted for publication in Proceedings B. Congratulations!!

Data Accessibility section

Open Access

Paper charges

Sincerely,

Dr John Hutchinson

Associate Editor:

Board Member: 1

Comments to Author:

Associate Editor: Doug Altshuler

The manuscript on the kinematics and energetics of fish swimming has now been seen by one of the reviewers from the previous round. They are fully satisfied with the revision and I agree. This is very interesting study with broad explanatory potential.

Appendix A

Response to Referee 1

The greatest concern I have is how the authors have presented the premise for the study. To my mind, the work successfully tests the hypothesis that speed change through frequency (and constant amplitude) is consistent with minimizing the cost of transport. However, there are passages in the abstract and introduction that give the misleading impression that the authors have assumed, rather than hypothesized, that fishes optimize swimming for efficiency.

Reply: Thank you very much for pointing out this misleading impression in the abstract and introduction. We have revised the abstract and introduction to clarify that it is a hypothesis that fishes optimize swimming for efficiency.

Specifically, in the abstract (Line 20-22), we state explicitly that “*Here we test the hypothesis that fish control tail-beat kinematics to optimize energetic expenditure during undulatory swimming.*” in the Introduction (Line 58–62), we state explicitly that “*Here, we examine if fish regulate tail-beat kinematics to optimize energetic expenditure during undulatory swimming. To test this hypothesis, we ask two questions: 1) Do body-wave frequency and swimming speed correlate due to energetic expenditure optimisation? 2) When optimising energetic expenditure at a given speed, do fish minimise cost of transport or maximise Froude efficiency?*”

Another concern has to do with the definition of the strategy that minimizes Froude efficiency in carangiform swimmers. As shown in Fig. 2b, the trajectory for that strategy lines the border of tail-beat amplitude values in the simulations. I am not clear on what limits that boundary, but regardless, I do not understand how it can be declared as a maximization without any simulations on one side of the trajectory. This issue does not occur in the other swimmers and it is clear that, the live fishes do not operate in that region. Nonetheless, I think there is an issue with the definition of the strategy in the carangiform case.

Reply: The trajectory of the strategy for Froude efficiency minimization was positioned at the right-hand-side border because of the restricted maximum body curvature and tailbeat amplitude that we considered in the simulations. This prevented us from plotting the actual trajectory, and the right border was used as a substitute. We apologize for this inaccuracy and the confusion that it has caused. In the revision, we have expanded the parameter space with new simulation that allowed us to find the “true” maximum Froude efficiency strategy trajectory. We have updated Figure 2 and the relevant description in the text.

On Referee 1’s specific comments

Abstract

It is debatable that animals optimize energetic performance during routine swimming. There are examples both in favor and in opposition to this statement among animals. I recommend leading with a less questionable statement.

Reply: We have changed the statement into “*Energetic expenditure is an important factor in animal locomotion.*”.

“which optimisation strategy”— Not sure what this means. If the authors are assuming optimizing energetic efficiency

and the strategy refers to how this is achieved, then there is a conceptual weakness b/c optimizing for something like efficiency should be a hypothesis and not an assumption. Please at least clarify what is meant by optimization strategy.

Reply: We thank the referee for pointing out that we used an unclear expression. “Optimisation strategy” means either maximising Froude efficiency or minimising CoT here, however, as you noted, it appeared too early in the text. We have revised this sentence to clarify. The new text is as follows:

“Here we test the hypothesis that fish control tail-beat kinematics to optimize energetic expenditure during undulatory swimming.”

What’s a “locomotory strategy”? Please define.

Reply: “Locomotory strategy” was an unclear expression for “energy-optimising strategy”, we have changed it into “energy-optimising strategy” .

“complexity of their kinematic optimisation:” is problematic on multiple fronts. “Complexity” is a tricky term in biology that can mean something that just has yet to be well-explained. Please be clear on the meaning of kinematic optimization. Is something about the kinematics optimized or do the kinematics optimize for some metric of performance. Is this the same thing as optimization strategy?

Reply: We agree that “complexity” reads confusing here in the abstract. “Complexity of their kinematic optimisation” meant “the number of dimensions of control-parameter space in fish swimming”—as tailbeat amplitude is kept stable, fish need only focus on tail-beat frequency, hence the control complexity is reduced.

Using “complexity” in the Abstract seems to insufficiently clear to readers. Thus in the revised version, we removed “complexity” from the abstract. Instead, in the discussion, we provide detail explanation that (Line 302-305) *“Hence, fish swimming control maybe simpler than previously thought: fish need not deal with a multi-parameter optimisation in the frequency–amplitude parameter space, but can simply focus on adjusting frequency, and can fix tail-beat amplitude near an ideal value. The almost linear relation between frequency and speed may further simplify speed control.”*

The science sounds exciting, but the language thus far is confusing and perhaps problematic, if it is assumed, rather than hypothesized, that fish move in an optimal manner.

Reply: Thank you for your suggestion. We have revised multiple phrases in the abstract to clarify that we hypothesise that fish move in an optimal manner is a hypothesis. Specifically, the first sentence in Abstract has been changed to *“Energetic expenditure is an important factor in animal locomotion.”* In Line 21–22, we state explicitly that *“Here we test the hypothesis that fish control tail-beat kinematics to optimize energetic expenditure during undulatory swimming.”*

Introduction

It’s a stylistic decision, but embedding equations in the middle of a sentence will unnecessarily alienate some of the broad Proc B readership. For example, one portion could be restated like this: “Analytical models show that thrust is proportional to the square of tail-beat amplitude and frequency, yet fish generally appear to control their speed by vary frequency.” It’s not really necessary to define terms like density when introducing the reader to the subject of this

study.

Reply: We agree with the referee. We have adopted the suggestion of the referee, and revised the other instances in the Introduction with an embedded equation in a sentence.

There are other metrics of “energetic performance”, such as the maximum power output. Perhaps it is more accurate to categories COT and Froude efficiency as metrics of efficiency.

Reply: Yes, it is a good point that “energetic performance” is inaccurate. However, considering that CoT may not be strictly called as an efficiency, we have revised “energetic performance” into “energetic expenditure” throughout the text.

“Is the correlation between body-wave frequency and swimming speed caused by energetic optimisation?” — It’s a bit odd to stat that energetic optimization causes a pattern of motion when the energetics are in fact caused by that motion. Perhaps rephrase with more biologically sensible causation.

Reply: We have revised this problematic sentence into “Do body-wave frequency and swimming speed correlate due to energetic expenditure optimisation? (Line 60–61.)”

“can be explained by one of two optimisation strategies (maximising η or minimising Ω)” — here I can see a clear meaning for “optimisation strategies”

Reply: Yes, the “optimisation strategies” is explained too late, therefore we have changed the expression of “optimisation strategies” prior to this explanation.

I like the approach outlined for this study. However, it is not the first to mix and match body shapes, Re, and kinematics (e.g., the work of Petros Koumoutsakos comes to mind: <https://doi.org/10.1017/jfm.2015.283>). An exhausted review is unrealistic for Proc. B, but some recognition of similar approaches in the literature would help place the current work in context.

Reply: We agree with the referee’s point. We have added two comprehensive numerical studies here, combined with specific hypotheses. Also, we further explain the difference in method between the current study and previous studies. The revised part is as follows:

“Combining these high-resolution performance maps with our extensive experimental dataset on larval fish allowed us to go beyond previous numerical studies [14,15] and test hypotheses about optimisation strategies used by actual fish.”

References:

14. Van Rees WM, Gazzola M, Koumoutsakos P. 2015 Optimal morphokinematics for undulatory swimmers at intermediate Reynolds numbers. *J. Fluid Mech.* **775**, 178–188. (doi:10.1017/jfm.2015.283)
15. Borazjani I, Sotiropoulos F. 2010 On the role of form and kinematics on the hydrodynamics of self-propelled body/caudal fin swimming. *J. Exp. Biol.* **213**, 89–107. (doi:10.1242/jeb.030932)

Methods

Please clarify if novel experiments were run as a component of this study or if the data were previously published. If the work is new, then the experimental methodology should be described in this paper or its supplemental materials. Citing other papers is insufficient.

Reply: Thank you. Experimental data from both published studies and new experiments was used in this study. We have clarified the source of experimental data, and added details of experiments in Section 2(a).

In Section 2(a), we state that “We used experimental data from published studies (figure 1, sources in Supplementary Materials) and new experiments (figure 4). We recorded cyclic swimming episodes of larval zebrafish at age 3 days post fertilisation (dpf) using a setup with three synchronized high-speed cameras described in more detail in previous publications [16,17]. The experimental setup can be found in Supplementary Materials. ”

In Ethics Statement, we added the information of the improvement for our animal experiment.

In Supplementary Materials E, we provide detail information of our experimental setting.

In Supplementary Materials F, we provide source of the experimental used in Fig.1.

(e) How was the optimization performed? On Page 8, there is mention of the value of performing an optimization, but it's unclear how the values of frequency and amplitude were arrived at.

Reply: Thank you for your advice. It is essential point. We have added a specific section “C-2 How to obtain optimal strategy curves” in Electronic Supplementary Materials to explain the process of optimization, and we now refer to this section in Methods section 2(e) in the main text. To support our verbal explanation we have added an infographic in figure S9.

Results

It seems to me that Eqn. 1 should be included in the Methods, not Results.

Reply: Agree. We have moved Eq.1 to the Method part, Section 2C.

P8 - I am similarly unclear on why the description of optimization methods (and Eqns. 2 and 3) are in Results.

Reply: We have moved the information corresponding to Eq.2 and 3 to the Method part, Section 2E, and changed their expression from equations to text.

Fig. 1b - Please plot the blue carangiform data in a manner that is visible. That data is currently hidden by the green points. I think it is important, b/c it looks like there may be amplitude modulation at low speeds.

Reply: Thank you for the suggestion. We have made the following improvement: on the one hand, we have shrunk the anguilliform points, so that both datasets are clearly visible; on the other hand, we have added three amplitude-vs-speed panels respectively for carangiform, anguilliform and larval swimmers to figure 1, showing the trends in all datasets.

Fig. 2 - I do not understand how the optimization for optimizing Froude efficiency can occur at the edge of parameter space tested. Isn't it possible that the global optima could be obtained at great values of tail amplitude? If that is not anatomically possible, then it still does not appear accurate to call that a strategy that maximizes the Froude efficiency.

Reply: We have implemented new simulations to expand the parameter space to identify the “true” maximum Froude efficiency strategy line. We have updated figure 2 and the relevant description in the text.

Eqn. 4 and associated text — again, it seems to me that this belongs in the Methods.

Reply: We have moved Eq. 4 to the Method part, Section 2C.

“larval fish as example” -> “larval fish as AN example”

Reply: This typo has been corrected.

(c) Why are thrust and drag calculations being shown? I do not believe there was a justification for this analysis prior to this section of the paper, so please explain what these calculations have to do with the paper's aims.

Reply: We appreciate this criticism, which helped us to improve the explanation in the paper. The thrust and drag calculations are needed to explain the underlying mechanism of our main findings.

- (1) To understand the drawbacks of Froude efficiency as an energetic indicator in undulatory swimming, we examined how body-wave kinematics affect thrust, drag and power. Our results demonstrate quantitatively a unique feature in undulatory propulsion – the undulatory wave is not only the source of thrust, but also impacts drag. Figure 5a, b and c suggest that, because the drag in undulatory swimming depends on swimming speed, and the nature of the undulatory kinematics, a swimming fish needs to optimise its undulatory kinematics to prevent an excessive energy expenditure due to drag. In contrast to the Ω_{\min} strategy, this requirement is missing in the maximum Froude efficiency strategy.
- (2) To understand the linear relationship between frequency and speed. We need to examine the relationship between thrust and frequency, and between drag and speed, and finally establish a linear relation between frequency and speed (at a constant tail-beat amplitude).

We noticed that the logical flow in our previous version was not sufficiently clear. Hence, in the revised version, we improved the logic of section 3(c) and (d), by adding sufficient explanations to guide the audiences, including the two abovementioned reasons. We also adjusted the order of the panels in figure 5 according to your advice.

Sections 3(c), (d) have been substantially changed, see the manuscript (revisions are highlighted in red colour).

P11 - Along the same lines, why are the fish being compared with spheres and plates? Please explain the justification for this comparison.

Reply: This issue is associated with the previous one. To understand why Froude efficiency may become an unreliable energetic indicator in undulatory swimming, we examined drag. Drag in undulatory swimming is not solely depending on swimming speed, but is also influenced by the undulatory kinematics. We found the (scaling) trend of drag follows that of a three-dimensional object (3D sphere and rigid fish body), while the magnitude

of drag depends on the kinematics of the undulatory wave, much exceeding that of a rigid fish gliding at the same speed. The 2D plate estimation has been used previously by the fish swimming community but we show that this is very inaccurate.

Therefore, spheres and plates are two important and necessary references to help to explain the (scaling) trend of fish drag, and to demonstrate the crucial role of body undulation in drag production. We have improved our expression in Section 3(c) accordingly.

Response to Referee 2

My main suggestion for the manuscript would be to incorporate more examples from living fish for the cost of transport. It would be interesting to know how the values obtained in this study match those of living fish. Cost of transport can be determined several ways from studies examining the metabolic rate of an animal at increasing speeds and it is not clear where the energy consumption in the equation for cost of transport in the supplementary information came from. Since the paper focuses on comparing cost of transport to Froude efficiency, more detail is needed on how cost of transport was obtained in the current study and how it relates to studies of living fish.

Reply: Thank you for the suggestion. To explain more clearly how cost of transport (as well as other parameters) was obtained, we have moved necessary explanations from the Supplementary Materials to newly created Section 2(e) in the main text. The section is now clearly presenting how we measured the forces and energetics, and refers to Section B-7 of Supplementary Materials for further details. It is important to note that in our approach, we only consider the mechanical costs rather than the metabolic costs. Nevertheless, we expect that reducing the mechanical costs will also reduce the required metabolic costs.

Concerning how cost of transport in the current study relates to studies of living fish, we have added a paragraph in the discussion to compare the mathematical and physical meaning between the two, and conclude that the optimal kinematics for the minimization of mechanical energy expenditure in this study can be applied to living fish, in spite of expected shape differences between of fluid dynamic speed-specific CoT trajectory between CFD and metabolic measurements as the latter include more components of the cost of transport than just the mechanical component considered in our ms. The paragraph is copied below:

“This study focused on fluid dynamic power, neglecting physiological contributions. Metabolic power is higher because the rate of energy expenditure at rest is not zero (basal metabolic power, P_{basal}) and locomotion involves the lossy conversion of chemical into mechanical energy. Yet the difference between the metabolic power and the basal rate, $P_{swimming}$, is positively correlated with mechanical power $P_{mechanical}$ [29,30], thus at a specific speed, minimising $P_{mechanical}$ is equivalent to minimising $P_{swimming}$, and the speed-specific Ω_{min} strategy obtained in this study may minimise speed-specific $\Omega_{metabolic}$. However, due to physiological contributions, the relation between metabolic power and speed is unlikely to be monotonic and may in fact be U-shaped [30,31]. In the future, our CFD approach could be combined with models representing the conversion of chemical energy into mechanical work by the swimming musculature.”

Reference

29. Webb PW. 1971 The swimming energetics of trout. J. Exp. Biol. 55, 521–540.
30. Gerry SP, Ellerby DJ. 2014 Resolving shifting patterns of muscle energy use in swimming fish. PLoS One 9. (doi:10.1371/journal.pone.0106030)
31. Di Santo V, Kenaley CP, Lauder G V. 2017 High postural costs and anaerobic metabolism during swimming support the hypothesis of a U-shaped metabolism–speed curve in fishes. Proc. Natl. Acad. Sci. U. S. A. 114, 13048–13053. (doi:10.1073/pnas.1715141114)

In terms of tail kinematics, Figure 1 is a nice example of the relationship of tail beat frequency in relation to size of fish and swimming speed, but it would be nice to see this relationship for tail beat amplitude. Since the authors suggest fish

should fix tail beat amplitude and adjust frequency to change speed, a figure of amplitude against speed for the species in Figure 1a would strengthen that result.

Reply: Thank you for this very useful suggestion. We have added new panels in figure 1c-e, to show the relationship between amplitude and speed.

Furthermore, it is mentioned in the manuscript that in the model tail beat amplitude is affected by tail beat frequency, but Figure 1b shows there is no relationship between the two. It is not clear if there should be a relationship and how this dependency in the model affects the results.

Reply: We apologize for our unclear explanation. The dependency between tail beat amplitude and tail beat frequency is a phenomenon only for simulations, rather than experiments. In simulation, we input the model fish tail motion by Eq.3 or 4 in a reference frame attached to the fish, while the output fish tail motion in a world reference frame was narrower, and influenced by tail-beat frequency. We have included a detailed explanation by Figure S10 and S11 in Supplementary Materials, and we now refer to these explanations in the figure captions in the main text.

My last main comment is to have more justification why a tetra was used as a model for a carangiform swimmer? Many of the species in Figure 1a are larger and the larvae was modeled after a zebrafish, so it wasn't clear why a smaller tetra species was used.

Reply: Thank you for the comment. We think the referee may agree that, from a dimensionless point-of-view, a tetra fish possesses general features of carangiform fish (the streamlined body and fan-shaped caudal fin) and looks very similar to other carangiform fish such as dace and carp. Hence, we think the referee's concern is about the body-size. Adult carangiform fish (as well as anguilliform swimmers) usually swim at $Re \sim O(>10^4)$, where turbulent flow occurs. It would be very challenging and time-consuming (in fact exceedingly hard to achieve with currently available hard- and software technology) to directly simulate the turbulent flow of carangiform and anguilliform fish in full scale, especially when tens of simulations were demanded in the current research. Meanwhile, previous studies suggest that key flow structure features in swimming seem robust to reducing Re , and there are many precedents that reduce the Re to $O(<10^3)$ in simulations of fish swimming (e.g. [24; Zhong et al, 2019]).

Therefore, we think it is necessary and reasonable to limit the body length of the carangiform and anguilliform swimmers to 20 mm to maintain Re within the laminar flow regime, to ensure the cost of computation is in a feasible range.

In the revised manuscript, we improved the explanation and validation on the usage of reduced-size carangiform and anguilliform swimmers.

Specifically, in Section 2(d) we state that *"We limited Reynolds numbers (Re) in this study to values from 1 to 6000 by capping the body length of swimmers at 20 mm. This limited Re range allowed us to calculate a large number of simulations at a previously validated grid resolution [23] with feasible time cost and without requiring a turbulence model, affording us high accuracy while maintaining key flow features that are robust against a reduction in Re [24]."*

Reference:

23. Li G, Müller UK, Van Leeuwen JL, Liu H. 2014 Escape trajectories are deflected when fish larvae

- intercept their own C-start wake. *J. R. Soc. Interface* 11. (doi:10.1098/rsif.2014.0848)
24. Liu G, Ren Y, Dong H, Akanyeti O, Liao JC, Lauder G V. 2017 Computational analysis of vortex dynamics and performance enhancement due to body-fin and fin-fin interactions in fish-like locomotion. *J. Fluid Mech.* 829, 65–88. (doi:10.1017/jfm.2017.533)
- Zhong Q, Dong H, Quinn DB. 2019 How dorsal fin sharpness affects swimming speed and economy. *J. Fluid Mech.* 878, 370–385. (doi:10.1017/jfm.2019.612)

Adding examples of smaller species to Figure 1a would be helpful for comparisons, or modeling a carangiform swimmer after a larger species would be an interesting comparison to see how the model holds up.

Reply: Following our logic in the previous reply, it is impractical to model a large carangiform swimmer with sufficient accuracy, while we totally agree making comparisons between our 20 mm model and smaller carangiform species is beneficial. In figure 1a, we have added a 5.5 cm long trout [Webb et al, 1984] and 6.9 mm long carp [Fu et al, 2013]. In the previous version of our ms, we already showed data of 5.2 cm and 9 cm dace [Bainbridge, 1958], and 4.5 cm jack mackerel, which is similar order as our simulation. Most important, we included also the most relevant experimental observation, tetra fish with a body length of 3.5~4 cm [Asharf et al, 2017]. We are confident that these data sufficiently demonstrate the statues for centimetre-level carangiform fish. We did our utmost to include additional datasets of small carangiform species, but these are so far the best data we managed to find.

In the revised version, we inserted a new panel in figure 1c, to clearly show the relationship between speed and amplitude. In figure 1b and c, we highlighted the data of tetra fish from other large carangiform swimmers; making comparisons with our CFD predicted amplitude, and the tetra fish data shows an amplitude around 0.16L, which is a better agreement with the predictions than those for larger carangiform swimmers. We also compared our results on carangiform swimmers with the average amplitude of a 9 cm dace in Bainbridge [Bainbridge, 1958], which was 0.153L, very similar to our prediction. (The data points we obtained for panel 1b and c are fewer than for panel 1a, because most papers show the frequency-speed relationship while much fewer literature present full frequency-amplitude-speed relationship.)

We have updated Figure 1, and added a comparison with small carangiform fish species in the main text, which is copied below.

“To explore which optimisation strategy comes closest to what actual fish do, we compared our predictions with published experimental data (references in Supplementary Materials). Amplitude values for dace (0.17L; 0.153L), trout (0.17L), and tetra fish (0.16L) are close to or within the predicted optimal range for Ω_{min} (0.14-0.16L) (figure 1b,c) but not for η_{max} (>0.25L) (figure 2b). ”

References:

- Bainbridge BYR. 1958 The Speed of Swimming of Fish as Related to Size and to the Frequency and Amplitude of the Tail Beat. *J. Exp. Biol.* **35**, 109–133.
- Ashraf I, Bradshaw H, Ha TT, Halloy J, Godoy-Diana R, Thiria B. 2017 Simple phalanx pattern leads to energy saving in cohesive fish schooling. *Proc. Natl. Acad. Sci. U. S. A.* **114**, 9599–9604. (doi:10.1073/pnas.1706503114)
- Webb, P. W., KostECKI, P. T., & Stevens ED. 1984 The Effect of Size and Swimming Speed on Locomotor Kinematics of Rainbow Trout. *J. Exp. Biol.* **109**, 77–95.

Fu C, Cao Z, Fu S. 2013 The effects of caudal fin loss and regeneration on the swimming performance of three cyprinid fish species with different swimming capacities. , 3164–3174. (doi:10.1242/jeb.084244)

Some minor comments:

In the methods, under (e): while terminal speed is defined, it is still a bit unclear.

Reply: Thank you for your advice, we have clarified the definition of speed measurement in Section 2(e). The new text is:

“In each simulation, the model fish accelerated from rest until thrust matches drag, resulting in an asymptotic increase in cycle-averaged swimming speed. We defined cyclic swimming as cycle-averaged swimming speed increasing by less than 1% from the previous cycle. All CFD results reported in this study were computed after the swimmer reached cyclic swimming.”

It is mentioned in the discussion that at low speeds, fish deviate further from the optimal amplitude than at high speeds, can you provide a reference?

Reply: We agree that this part needed further clarification. We provide examples of this deviation-- in figure 1b and c, the data points at the lowest speed side drop sharply, while our CFD predictions consistently suggest stable tail-beat amplitudes. We further provide additional discussion on the possible reasons of such deviation.

“At low swimming speeds, experimental observations usually show a smaller tail-beat amplitude (figure 1b and c, data points at the lowest speed side drop sharply) than predicted by CFD. However, this deviation results in a very low energy penalty because both power output and Ω are very low at these speeds, and there may exist some physiological factors (for instance muscle fibre types) that prevent fish from using low-frequency-large-amplitude kinematics.”

Appendix B

Response to Referees (including a copy of manuscript with ‘tracked changes’)

Response to Referee: 3

General comments:

[Point 1] Generally, I question scaling by body length, especially when body shapes vary. The authors will probably do this anyway, and as it is consistent with the vast majority of the literature, I am willing to let it slide.

Reply: Thank you for sharing your concern about scaling. We agree that scaling requires care in application and interpretation. As the reviewer points out, it facilitates direct comparison with the literature. Scaling necessarily forces us to make choices about which feature becomes the ‘characteristic’ length and hence cannot represent all morphological variation, yet such simplifications are unavoidable when aiming to discover general principles across species and scales. We hope that our explanation addresses the reviewer’s query.

[Point 2] The phrasing of the main questions in the introduction seem to assume a priori some kind of optimization strategy (the two being studied as contenders), when in reality, it is theoretically possible for the fish not to be optimizing anything.

Reply: Thank you for sharing your concern that this phrase suggests a Panglossian fallacy. We agree that it is possible that fish are not optimizing anything. Our phrasing follows the format of a scientific hypothesis (observation plus proposed explanation) and was chosen in this manner intentionally to imply the null hypothesis that fish do not optimise energetics. After careful consideration, we opted to leave the original phrasing because it most closely accords with so-called classic writing style, as recommended by Steven Pinker (a scholar in linguistics and expert of scientific writing). Please refer to Point 5 for details.

[Point 3] I think the authors should weaken their claim that these findings apply to all fishes. Given the Reynolds regime studied, and the lack of extensive data backing up the assumption that laminar flow regimes are generally similar to turbulent flow regimes in fishes, I think some discussion of this limitation is appropriate.

[Point 6 (associated with Point 3)] Ln 74: It is a bit strong to suggest that the data in this study apply to all fish when the size of fishes tested was only 2cm, when that is still in a laminar regime—though I understand the complications involved with simulating turbulent flows.

Reply: Thank you for sharing your concerns about overgeneralizations. First, this study does not claim that “these findings apply to all fishes”. We do imply, however, that our findings are generalizable

(apply to more fish than the ones we collected data on). The statement “fish swim” is not logically equivalent to “all fish swim”. Second, the queried line in the manuscript states that “this strategy successfully predicts observed behavior”, that is: we claim that our simulations, done in the laminar flow regime, successfully predict experimental observations (a correlation between tailbeat frequency and swimming speed). Given that this correlation has been observed in a wide range of fish, this statement does indeed imply that our findings on fish swimming the laminar flow regime might also apply to fish swimming the turbulent flow regime. Such an implication does not constitute an overinterpretation. To address this concern, we reread the manuscript carefully to ensure that there are no categorical claims along the lines of ‘all fish’ in the text.

Specific comments:

[Point 4 (associated with Point 1)] Ln 49: I understand the desire to make things unitless and normalize for body size, but I generally question the strategy of doing this by dividing by body length. Yes, there is precedent for it in the literature, but when you are dealing with fishes of radically different shapes/aspect ratios, does it make sense to do this? Is dividing by body length for a 2cm eel the same as dividing by body length for a 2cm tetra?

Reply: Thank you for sharing your concern. We agree that shape is important. This is why we simulated three different body shapes and their corresponding body wave kinematics. We use non-dimensionalised lengths (by normalizing to body length) to report length values, in particular tail beat amplitude. This normalization does not lead to our study neglecting shape differences between fish, it merely removes absolute units to facilitate comparison across studies and to comply with professional standards about reporting such data. We hope that our explanation addresses the reviewer’s query.

[Point 5 (associated with Point 2)] Ln 64-67: It seems to me that, if you are not assuming energy optimization outright (which I hope the authors are not), question 2 is 100% predicated on the answer to question 1 being “yes”. The way question 2 is phrased also sounds like fishes MUST be optimizing in one of those two ways... even though, hypothetically, both could be false. I would suggest reframing these questions to avoid upfront assumptions about optimization, and allow for the possibility of other explanations. For question 2, the word “if” would go a long way.

Reply: Thank you for sharing your concern about our study aim. We propose energy optimization as a possible explanation for the observed correlation between tailbeat frequency and swimming speed (scientific hypothesis). We then test this hypothesis by comparing how well two widely used energetic measures (Froude efficiency, cost of transport) predict observed behavior. One possible outcome of this study is indeed that neither measure predicts observation, in which case we would

need to reject our hypothesis. To address this concern, we implemented the reviewer's suggestion concerning the second question.

[Point 7 (associated with Point 2 & 5)] Ln 75: Sounds like there is a correlation/causation issue here. The data in this study support that fishes are minimizing mechanical cost of transport, but they do not exclude the possibility that fishes are optimizing something else (metabolic CoT). This is mentioned in the discussion, but perhaps could be mentioned earlier.

Reply: Thank you for sharing your concern about causation. Our study is proposing a mechanistic explanation for an observed correlation, by detailing the fluid-dynamic mechanisms that would cause the observed pattern. No mechanistic explanation excludes other, additional or alternative, explanations, but the principle of parsimony applies. To address this concern, we have revised the last sentence in Introduction. *"We show that fish can control swimming speed by changing frequency rather than amplitude to minimise fluid-dynamic speed-specific cost of transport rather than maximise Froude efficiency, and that this strategy successfully predicts experimentally observed behaviours."*

[Point 8] Ln 83 and throughout: The supplementary materials are extensive (and very clear. Thanks!!!). It would be incredibly useful to the reader if the authors indicate which section of the supplement they are referring to at a given place in the manuscript.

Reply: Thank you. To implement this suggestion, we added section numbers of supplementary materials throughout the manuscript.

[Point 9] Ln 95: Do tetra swim using carangiform kinematics? At least in aquaria, they seem to be intermittent swimmers. It would be useful to see a reference for this, if one exists.

Reply: Thank you for your query about Tetra. Our study focuses on the effects of body wave shape and body shape, not the effects of cyclic versus intermittent swimming. Intermittent swimming is indeed used by many fish, and it is the dominant mode in adult Tetra, yet its analysis is beyond the scope of this study. In the context of this study, it is relevant that Tetras are carangiform swimmers; see e.g. Hebrank (1982): *"Typical carangiform swimmers include the carangids for which this style is named, such as the pompano (Trachinotus carolinus) as well as the characins (popular tropical aquarium fishes) such as the tetras."* To address this concern, we would like to emphasise that we included several references that define carangiform swimming (Sfakiotakis et al. 1999; Wardle et al. 1995).

References

Hebrank, M.R. (1982). Mechanical properties of fish backbones in lateral bending and in tension. *Journal of Biomechanics*, 15(2), 85–89. doi:10.1016/0021-9290(82)90039-2.

Sfakiotakis, M., Lane, D.M., & Davies, J.B.C. (1999). Review of fish swimming modes for aquatic locomotion. *IEEE Journal of Oceanic Engineering*, 24(2), 237-252.

Wardle, C.S.J.J., Videler, J., & Altringham, J. (1995). Tuning in to fish swimming waves: body form, swimming mode and muscle function. *Journal of Experimental Biology*, 198(8), 1629-1636.

[Point 10] Ln 98-100: Were these kinematics, or the resulting kinematics from the simulations, compared to those of actual fishes swimming? Would be useful to know how near a match they are.

Reply: Thank you for your query about the accuracy of our simulations. This study and others (Borazjani and Sotiropoulos, 2009) simulate body kinematics that mirror experimental observations as well as counterfactual cases to examine how kinematics affects performance. By comparing experimentally observed performance with simulations, we can not only confirm that simulations using experimentally observed body kinematics correctly predict swimming kinematics (such as swimming speed), but we can also test hypotheses about the effect of kinematic parameters (frequency, amplitude) on hydrodynamic performance. So in answer to this query, this study and others (Li et al. 2012, Borazjani and Sotiropoulos, 2009) have in fact shown that CFD simulations using experimentally observed kinematics can accurately predict swimming performance.

References

Li, G., Müller, U.K., van Leeuwen, J.L., & Liu, H. (2012). Body dynamics and hydrodynamics of swimming fish larvae: a computational study. *Journal of Experimental Biology*, 215(22), 4015-4033.

[Point 11] Ln 121-122: Again, I am wondering how robust these findings are to changes in body size. The reference provided here, Liu et al 2017, does not itself provide evidence that this is the case, though it contains a discussion of literature. One of the studies cited as evidence for the similarity of laminar vs turbulent flow fields (Müller et al. 2001) does not appear to directly address this—if it does provide evidence of this similarity, it is not clear. Furthermore, Müller 2001 only generated 2D flow visualizations, making it unsuitable for comparison to a 3D simulation. The reference Müller 2001 is being compared to (Kern & Koumoutsakos) is only a simulation for the anguilliform case. All of this is to say that, while I understand using low Re simulations for computational feasibility, in the absence of more data, I am less certain of their application to

larger fish. This limitation should at least be mentioned and discussed, rather than dismissed out of hand. Unless I am grossly misinterpreting these studies.

Reply: Thank you for sharing your concern. It is correct that due to the computational feasibility, most computational research (including this study) limits Re at 10^3 . To address these queries, we firstly now refer to Liu et al. 2017, who state that “*previous studies (Buchholz & Smits 2006; Bozkurtas et al. 2009) have shown that the major flow features in flapping propulsion are similar for a changing Reynolds number.*” Secondly, concerning the comparison between Kern & Koumoutsakos and Müller et al. 2001, we would like to point out that a comparison of two-dimensional section through 3d simulations with two-dimensional sections through a three-dimensional experimental flows is in fact valid. Kern & Koumoutsakos simulated a fish swimming at Re 2400 to 3900, which 5 to 30 times lower than the Re in the experimental studies, yet the simulated wake patterns showed strong similarities with the experimental results (Müller et al. 2001). Without simulations in the turbulent regime, all we can currently say is that available evidence suggests a similarity in major flow features, but we lack data to make such a statement about the full characteristics.

To address this concern, we revised our manuscript in the Introduction to state explicitly that our simulations suggest that the proposed optimisation strategy “*successfully predicts experimentally observed behaviours in both the laminar and turbulent flow regime.*” In the Discussion, the revised text reads “*Simulations in laminar flow regime on carangiform and anguilliform swimmers generate similar results to those of larval fish, and correctly predict the observed close correlation between tailbeat frequency and swimming speed. Although our simulations are limited to the laminar flow regime, key flow features appear to be robust across flow regimes [Liu et al. 2017], and the predicted trends occur also in fish swimming in the turbulent flow regime, suggesting that undulatory swimmers adopt the Ω_{min} strategy over a wide range of size and developmental stage.*”

References

- Liu G., Ren Y., Dong H., Akanyeti O., Liao J.C., Lauder G.V. 2017 Computational analysis of vortex dynamics and performance enhancement due to body-fin and fin-fin interactions in fish-like locomotion. *Journal of Fluid Mechanics*. 829, 65–88.
- Buchholz, J.H., & Smits, A.J. (2006). On the evolution of the wake structure produced by a low-aspect-ratio pitching panel. *Journal of Fluid Mechanics*, 546, 433-443.
- Bozkurtas, M., Mittal, R., Dong, H., Lauder, G.V., & Madden, P. (2009). Low-dimensional models and performance scaling of a highly deformable fish pectoral fin. *Journal of Fluid Mechanics*, 631, 311-342.
- Kerm, S. & Koumoutsakos, P. (2006). Simulations of optimized anguilliform swimming. *Journal of Experimental Biology* 209 (24), 4841–4857.

Müller, U.K., Smit, J., Stamhuis, E. J., & Videler, J.J. (2001). How the body contributes to the wake in undulatory fish swimming: flow fields of a swimming eel (*Anguilla anguilla*). *Journal of Experimental Biology*, 204(16), 2751-2762.

[Point 12] Ln 178-189: Looking back at the figures showing the performance surfaces for CoT and Froude efficiency, it looks like the combinations of frequency and amplitude that are good for CoT minimizing are explicitly bad for Froude maximization and vice versa. Is there a reason to a priori expect there to be a tradeoff between the two? If yes, it would be great to see that in the introduction and/or the discussion.

Reply: Thank you for sharing this idea. We would not go as far as claiming that the two strategies are opposing each other, implying that they act on the same aspect of hydrodynamic performance, but push it in opposite directions. Instead, the two strategies “*differ in which aspect of hydrodynamic performance is optimised, which leads to their optima to be situated at different locations in the kinematics landscape. Optimising η requires maximising the ratio of useful power to power expended on the water, irrespective of net energetic expenditure per unit distance. In contrast, optimising Ω requires minimising input power to move a given mass at a given speed.*” We hope that this explanation addresses the reviewer’s idea.

[Point 13] Ln 200-205: It would be useful to see a depiction of different values of beta.

Reply: We are not confident that we fully understand this comment. We understand your comment as ‘a depiction is needed to better explain the physical meaning of the different values of beta’. Based on this understanding, we have added a legend in figure 5, which provides intuitive demonstration of different values of beta. We hope that this explanation addresses the reviewer’s concern effectively.

[Point 14] Ln 231-232: How were these “physiologically realistic limits” determined?

Reply: Thank you for sharing your concern. We limited the upper ranges near known tail beat frequencies and tail beat amplitudes for each type of swimmer. To address this concern, we have revised the text as follows “*We estimate the theoretically possible space by varying body kinematics ... within biologically relevant limits.*”

[Point 15] Ln 238: I find the use of equations in parentheses in sentences, especially when two equations for opposite things are in series like this, to be quite difficult to parse.

Reply: Thank you for sharing your concern. We revised the text as follows “As Re increases from 10^0 to 10^3 , the relation between body drag and speed changes from $D \propto U$ (i.e., $C_D \propto Re^{-1}$ at $Re \sim O(<10^0)$) to $D \propto U^2$ (i.e., $C_D \rightarrow \text{constant}$ at $Re \sim O(>10^3)$) (figure 5b). Because time-averaged drag and thrust match during cyclic swimming ($T=D$), thrust is approximately proportional to f^2 ($T \propto f^2$), and $D \propto U^2$ at $Re \sim O(>10^3)$, the relation between f and U changes to $U \propto f$ at high speeds.”

[Point 16] Ln 245-250: The experimental data look like there is a curvilinear (maybe quadratic?) relationship between w and U , as opposed to a linear one. Is there an explanation for that? I think what I am saying is I disagree with the statement that U is roughly proportional with w .

Reply: Thank you for sharing your thoughts. The text expressly uses the qualifiers “roughly” and “across a wide size range” to indicate that this relationship is not strictly linear across the available data set. The manuscript contains several sections that explain when and why the relation between w and U is not strictly proportional.

In the Methods (section 2d, second paragraph), we point out that “because time-averaged drag and thrust match during cyclic swimming ($T=D$), thrust is approximately proportional to f^2 ($T \propto f^2$), and $D \propto U^2$ at $Re \sim O(>10^3)$, the relation between f and U changes to $U \propto f$ at high speeds.” Thus when the speed (and body size) is low enough for fish to leave the inertial flow regime, the $U \propto f$ approximation no longer applies and the slope changes. As shown in Figure 1e, the minimal CoT curve and all iso-curvature-amplitude curves (those blue bead-chains) all have a flat low-speed region and appear curvilinear. This is to say, our optimal strategy already predicts a curvilinear low speed region.

In the Discussion (fourth paragraph), we address a second factor, the actual behavior of fish at low speed: “At low swimming speeds, experimental observations usually show a smaller tail-beat amplitude (figure 1b and c, data points at the lowest speed side drop sharply) than predicted by CFD. However, this deviation results in a very low energy penalty because both power output and Ω are very low at these speeds, and there may exist some physiological factors (for instance muscle fibre types) that prevent fish from using low-frequency-large-amplitude kinematics.” Fish tend to use a smaller tail beat amplitude at lower speed, causing the experimental data to deviate from the predicted curves at low speed.

To address this concern, we revised the text to “Comparison with experiments shows that U is indeed roughly proportional to w across a wide speed- and body-size range.”

[Point 17] Ln 256-258: I find a lot of things about this sentence’s phrasing troubling. I highly doubt that fish operate at a “given” frequency, and they certainly are not “choosing” an amplitude to

“maximize speed”. If anything, the fish “chooses” a speed, and modulates frequency. The order of operations here does not make sense. I am not sure I understand how these conclusions are being drawn, but that may be a limitation of my own knowledge. That said, for a Proc B readership, perhaps this should be explained more clearly.

Reply: Thank you for your sharing your concern. To address this concern, we revised the text as follows.
“At a given frequency within this parameter space, the fish does not operate at the amplitude that maximises speed, but at a lower amplitude close to the Ω_{min} trajectory (figure 4b).”

[Point 18] Ln 272-292: I love these paragraphs. But, in line 287, I wonder about the use of these references to support the claim that $P_{mechanical}$ and $P_{metabolic}$ are highly correlated. It is a very reasonable assumption – but neither of these studies measured mechanical power output. So, I would say that it is a safe assumption, but not known.

Reply: Thank you for sharing your thoughts. The cited references show that swimming speed correlates with oxygen consumption. To make the link between oxygen consumption and swimming power more explicit, we revised the text as follows *“Yet $P_{swimming}$ (difference between metabolic power and basal metabolic power) increases with swimming speed [29,30] and therefore positively correlates with mechanical power $P_{mechanical}$.”*

[Point 19] Ln 303-304: Can the authors (very briefly) explain how muscle fiber types would preclude low-frequency large-amplitude kinematics? I can come up with an explanation, but the reader may not.

Reply: Thank you for sharing your concern. Given the strict length limit imposed on manuscripts by Proceedings B, we decided after careful consideration to address this concern by making the statement more general rather than more specific. The revised text reads *“However, this deviation results in a negligible energy penalty because both power output and Ω are very low at these speeds, and there may exist physiological factors (such as muscle physiology) that prevent fish from using low-frequency-large-amplitude kinematics.”*

[Point 20] Figure 1a: I have a very hard time distinguishing the colors of the eel, trout and dace, and the jack mackerel, carp and dace points. Can the authors maybe use different shapes in addition to color?

Reply: Following your advice, we have redrawn figure 1a.

[Point 21] Figures 2-4: This may just be me being picky, but in general I consider brighter colors to be “better”. For the c,d,e panels, this results in a different frame of reference: while yellow agrees with my expectation in c and e, d is the opposite of what I expect. I spent a lot of time completely misinterpreting the d panels because of the color scheme. Maybe the authors could flip the color palette for panel d? But this may not be a concern for other people.

Reply: Thank you for sharing your thoughts. In the panels of figures 2, 3 and 4, we consistently use “hot (bright)” colours for large algebraic values and “cold (dark)” colours for low algebraic values. This practice is widely accepted in the natural sciences. Therefore, after careful consideration, we kept the current design.

[Point 22 (associated with Point 8)] Supplemental materials: well organized, well presented and very helpful. I do wish some of these were included in the main text, but I understand space limitations here.

Reply: Thank you for your sharing your thoughts. To meet the journal requirements, we have moved much background information into the Supplemental Materials. Yet the manuscript in the current form is still beyond the standard length (we have already extended it from 6 to 10 pages). To facilitate the reader’s access and use of the supplementary materials, we followed your advice and added section numbers wherever we refer to Supplemental Materials.

Response to Referee: 4

The question presented is central and timely in fish biology. More broadly, whether and how animals optimize their morphology and behavior is of broad interest. I am a fan of the approach presented in this paper, which combines numerical simulations and observations. Therefore, this study can be of interest to the broad readership of ProcB, but some of the parts will have to be re-written to be suitable for this readership.

[Point 23] Specifically, the intro is inaccessible to non-experts (as is immediately apparent with the use of technical terms such as “dimensionless tail-beat amplitude” in the first paragraph). The introduction must provide the general reader some sort of intuition about the problem, and gradually introduce them to the technical terms such as “cycle averages of respectively thrust, swimming speed, and input power”.

Reply: Thank you for sharing your concerns about readability. To address them, we made the following changes. We replaced the term ‘*dimensionless amplitude*’ with a definition ‘*tail-beat amplitude A (normalised amplitude, i.e. expressed in units of body length)*’, and we replaced the term ‘*cycle-averaged*’ with the phrase ‘*averaged over one tail beat cycle*’.

[Point 24] Also, the paper dwells quickly into the question of which metric of optimization is more important, but fails to explain why this is a general and interesting question in biology (and it is!).

Reply: Thank you for sharing your concern about context. As the reviewer notes in their next comment, this manuscript operates under extreme space constraints. We therefore quickly progress from ‘observation’ (correlation between speed and frequency) to hypothesis (energetic optimisation). To address this concern, we revised the text as follows “*With energetics being an important factor during routine activities, fish might change frequency rather than amplitude to optimise energetic expenditure during cyclic swimming.*”

[Point 25] Another issue with the introduction is that it glosses over previous attempts of dealing with this question, some of which were carried out by authors of this study but also by others. There should be an explanation of why this study is needed in the light of these previous attempts. I understand that there is a word limitation in this journal, but I see these issues as essential.

Reply: Thank you for sharing your concerns about providing context. Rather than stating in more detail the merits and shortcomings of previous studies, we opted to address this concern by highlighting what makes this study unique—the combination of experimental and computational data allows us to actually test energetics hypotheses by placing experimental observations in predicted performance landscapes, which allows us to compute the difference between observed and

predicted behavior, and quantify which optimization strategy predicts observed behavior better. To address this concern, we revised the text as follows “*Combining these high-resolution performance maps with our extensive experimental dataset on larval fish allowed us to go beyond previous numerical studies [14,15] and actually test hypotheses about optimisation strategies used by actual fish.*”

[Point 26] A second issue is related to the limits of interpretation. In general, the parameter ranges for the modeled and observed data only partially overlap. This limitation is not discussed, and comparisons are made irrespective of it. This seems like a major problem in the design of the study and it should be amended. Either expand the performance maps to match the range of observed data, or collect data that would fit the current maps.

Below I expand on this point:

For all the right reasons, numerical simulations were carried out for a range of Reynolds numbers of <2000 , limiting body size to 20 mm. However, except for larval zebrafish, none of the fish for which data is provided swims at such low Re/small size. Thus, most of the data presented as evidence for the fit between the numerical solution and the observations is outside the parameter range of the simulation and is therefore a gross extrapolation. Fig 1 clearly shows that size is a major determinant of frequency range, but this is ignored in the interpretation of the results. Given that collecting data on the frequency and amplitude of swimming fishes is not a highly demanding task, the immediate solution to this would be to obtain experimental data that fits the parameter range of the simulations. Even after this essential addition, the authors must acknowledge the limitations of their data, and avoid over-interpretation to domains outside of their parameter range. That is, applying the results to “all fish” warrants at least some caution.

Reply: Thank you for sharing your concerns about overgeneralizations. Let us first point out that this manuscript does not claim to apply to “all fish”. We do imply, however, that our findings are generalizable (apply to more fish than the ones we collected data on). The statement “fish swim” is not logically equivalent to “all fish swim”. Second, in our opinion, addressing this issue requires not so much additional experimental data on small fish but simulations in the turbulent flow regime. In the absence of such simulations (which are simply not feasible at this moment: increasing the scale of carangiform and anguilliform swimmers requires turbulence simulation, which are impractical with current technology for parametrical investigations that require several tens of simulations), we phrase carefully to avoid overinterpretation. We provide a thorough validation for the case for which we have experimental data. Beyond the range of our experimental data, we claim that our simulations, which were done in the laminar flow regime, successfully predict experimental observations (a correlation between tailbeat frequency and swimming speed) in the inertial flow regime. This statement does indeed imply that our findings on fish swimming in the laminar flow regime might (not ‘do’) also apply to fish swimming in the turbulent flow regime. This implication does not constitute an overinterpretation of our data. To address this

concern, we reread the manuscript carefully to ensure that we did not unintentionally make categorical claims in the text, but could not find any, such as “all fish” cited by the reviewer.

Reference

Liu G., Ren Y., Dong H., Akanyeti O., Liao J.C., Lauder G.V. (2017). Computational analysis of vortex dynamics and performance enhancement due to body-fin and fin-fin interactions in fish-like locomotion. *Journal of Fluid Mechanics*, 829, 65–88.

[Point 27] Along the same lines, what determined the range of relative speeds modeled for the different swimmers (Fig 2-4)? Was there any attempt to match the range to the parameter range observed in nature? A superficial examination of Fig 1C reveals that the modeled range for anguilliform exceeds that presented in the data by 2-fold while that for carangiforms is narrower by 50%. Specifically, $\frac{3}{4}$ of the points for Tetra fall outside of the modeled line. How is this justified? Surely this will affect interpretation/confidence?

Reply: Thank you for sharing your thoughts. We would like to make two points. First, concerning our choice of limits in relation to the aims of this study: for the larval fish, the exact limits of the range are not relevant as long as the limits are wide enough to create a performance landscape that allows the experimental data points to fall within the simulated landscape rather than outside of it. For the carangiform and anguilliform swimmer, the landscapes should overlap with that of the larval fish to demonstrate that the shape of the landscape for a larval body shape and wave shape is similar to that of a carangiform and anguilliform body and wave shape, so we can safely propose that the findings for larval fish (which are experimentally verified) might be generalizable to other body shapes and body wave shapes. Second, concerning our choice of limit in relation to biologically observed values: we chose frequency and amplitude values for the simulations that covered most of the typical value range for carangiform and anguilliform swimmers. Frequency, amplitude control factor, and curvature control factor are input variables; frequency ranges from 2 to 18 Hz; the amplitude control factor ranges from 0.02 to 0.18 for anguilliform and 0.22 for carangiform swimmers; so we can confirm that the modelled range indeed matches the range of values that has been observed experimentally in these types of swimmers. Tail beat amplitude and swimming speed are output variables (our CFD models free-swimming fish, so the amplitude is a result of the body wave interacting with the fluid), and hence their ranges differ more between anguilliform and carangiform swimmers. In summary, the choice of upper limits for the input values was driven mainly by the aim to generate a performance landscape that is wide enough to display the two optimization strategies and to allow a comparison with the larval performance landscape. The exact limits are therefore somewhat arbitrary. By how much the simulated landscapes differ in size does not affect the interpretation or confidence; what matters is that the landscapes are large enough to show include the lines that describe the speed-specific optima for

cost of transport and Froude efficiency. We hope that these explanations address the reviewer's questions.

There are several additional issues that need to be addressed in revision this paper, mostly related to the relevance of the parameter range of the computational work to the real-world data: **[Point 28]** 1) The range of relative speeds in the carangiform and anguilliform swimmers compared to the larval fish is perplexing. Surely larval zebrafish are not the fastest fish in the world (in terms of BL/s)? if the speed of the other fish is capped due to computational limitations, than this again dampens the ability to really understand which metric is optimized at relative speeds that are relevant to the real world.

Reply: To be honest, we are as perplexed by the reviewer's query as the reviewer is by our data. By way of an answer, we would like to first explain the conceptual framework of this study, then address the fact that larval fish reach high length-specific speeds.

First, this study compares predicted performance landscapes with experimental data for larval fish to test the hypothesis that larval fish use body wave kinematics that optimises hydrodynamic energetics (minimise cost of transport or maximise Froude efficiency). The study then expands this work on larval fish (simulations using experimentally observed body shape and body wave shape) to other swimmers (carangiform and anguilliform body shape plus wave shape) to examine how sensitive the performance landscape predicted for larval fish is to body and body wave type. Due to computational limitations, those simulations are done in the laminar flow regime. We found that the three types of swimmers show similar trends. The starting point of this study was that there is a generally observed correlation between tailbeat frequency and swimming speed in undulatory swimmers across body size and exact type of undulatory swimming style (from anguilliform to tuniform). Given the similarity in predicted performance landscapes, this study proposes that the experimentally supported conclusions concerning larval fish might also apply to swimmers with other body shapes and body waves, which amounts the suggestion that our findings might be relevant also to swimmers in the turbulent flow regime.

Second, in answer to the reviewer's more specific concern about range differences in swimming speed. Small organisms, including larval fish, move at higher speeds (when speed is expressed in body lengths) than large organisms. This fact has nothing to do with 'capping' speed due to computational limitations. So this study does not cap the speed of large organisms to values that are unrealistically low and hence biologically not relevant. Instead, this study shows that an experimentally observed correlation between tailbeat frequency and swimming speed can be explained by hydrodynamic optimisation; we show this for larval fish in the laminar flow regime, and we propose that this finding might apply also to other swimmers and in the turbulent flow regime. See also our reply to Point 29.

We hope that these explanations address the reviewer's questions.

[Point 29] 2) Can larval zebrafish be classified on the anguilliform-carangiform spectrum? Also, the range of speeds given here is ~2-5 fold greater than reported in other studies of zebrafish swimmers. can you explain this?

Reply: Thank you for sharing your query. Yes, larval zebrafish can be classified on the anguilliform-carangiform spectrum as follows. Larval fish, Tetras and eels are all undulatory swimmers. Undulatory swimmers can be further sub-divided based on the shape of their body wave, mainly body wave length and the shape of the amplitude envelope (for a review see Wardle et al., 1995). Based on their body wave length and amplitude envelope, zebrafish larvae more closely align with anguilliform than carangiform swimmers; yet larval fish differ from adult undulatory swimmers in ways that might have to do with their body shape, developmental stage, and flow regime. Concerning the values for length-specific speed, the values reported here are larger than that of adult zebrafish but consistent with values reported in other studies of larval zebrafish (e.g. van Leeuwen et al. 2015; Voesebeck et al. 2020). Yes, we report some record-speed episodes that were part of escape responses, where the fish were startled and responded by performing a C start followed by an extended period of near-cyclic swimming. We hope that these explanations address the reviewer's questions.

References

- van Leeuwen J.L., Voesebeck C.J., Müller U.K. (2015). How body torque and Strouhal number change with swimming speed and developmental stage in larval zebrafish. *Journal of the Royal Society Interface*, 12, 20150479.
- Voesebeck C.J., Li G., Muijres F.T., van Leeuwen J.L. (2020) Experimental–numerical method for calculating bending moments in swimming fish shows that fish larvae control undulatory swimming with simple actuation. *PLOS Biology* 18, e3000462.

[Point 30] 3) Fig 1 is beautiful but can be confusing. It is unclear what are the gray lines are, and why some genus appear twice (e.g. trout) while others appear once. Panels a and b are redundant- pane B should be in the ESM. Panel e (and also d to a lesser extent) does not really belong to the panel.

Reply: Thank you for your query. 1) The gray lines connect the data points to the experimental fish icons plus text indicating their length. Some fish species have multiple data sets and therefore may appear multiple times if those data sets have widely different slopes (e.g. jack mackerel, trout). If the datasets belonging to the same species have similar slopes (e.g. dace), we combined them to

keep the figure concise. To address concerns about the figure, we redrew figure 1a to more clearly shown the data sets and fish.

2) Panels (a), (b) and (c) show speed-frequency, amplitude-frequency and amplitude-speed relationships. Panel (b) is not redundant with panel (a) because the two panels show different relationships, as evident by the axes labels.

3) We combined panels (d) and (e) with panel (a) to (c) mainly to meet the formatting requirements of Proceedings B, which limits us to five figures. Given that panel (d) re-uses the data from panel a and panel e needs to be compared with panel (d), figure 1 is the most suitable figure to contain panels (d) and (e). We revised the caption for panel (e) to clarify and emphasise its source data are in figure 4.

We hope that these explanations address the reviewer's concerns.

[Point 31] 4) Panels c2 and c3 give the impression that speed and frequency are correlated (BTW, why are the axes flipped compared to panel a?). What would be the results of a multiple regression model that took into account the effects of amplitude and frequency (and perhaps size) on speed? I am guessing all three will be significant.

Reply: Thank you for your suggestions. Panels (c2) and (c3) show amplitude, not frequency, as a function of speed, so there is no direct flipping of axes. To explain in more detail: We opted to use frequency as the x axis for panel (a), and therefore also panel (b). This makes amplitude along the y axis for panel (b). To keep things consistent between panel (b) and panels (c1,2,3), we kept amplitude as the y axis in panels (c1,2,3), making speed the x axis in panels (c1,2,3).

Concerning a multiple regression analysis, this study opted for a mechanistic rather than a statistical analysis of the relation between amplitude, frequency, and swimming speed.

We hope that these explanations address the reviewer's questions.

[Point 32] 5) Denoting the ± 5 and 10% lines for the minimum-cost-of-transport curve in Fig 2-4 is essential, but why is are the same lines omitted for Froude efficiency?

Reply: Thank you for the suggestion to add those lines. After careful consideration, we decided to include the lines in a figure in the supplementary materials (ESM Fig. S13), but not the main manuscript. This decision reflects that (1) the maximal Froude efficiency strategy was rejected and (2) adding four more curves into the already complex panel (b) may be more confusing than helpful.

[Point 33] 6) The explanation of how the curves for minimum-cost-of-transport and Froude efficiency are drawn should be in the paper and not in the (excellent and detailed) ESM file.

Reply: Thank you for your suggestion. Proc. B limits paper length and the number of figures, and we have used up all available space. We believe putting “how the two curves are made” in ESM file can provide more room for a clear explanation, whereas placing this information in the main text would force us to severely abridge those explanations. To facilitate access, the main text now gives exact section references to the ESM file.

[Point 34] 7) Also, why does the calculation of the two curves take into account speeds? I am not sure that I understand this from the explanation in lines 55-60. Both metrics have U in them already?

Reply: Thank you for your query. Our study distinguishes between global optimum and speed-specific optima. For this explanation, we will focus on cost of transport as the optimisation strategy. The performance landscape shows that there is a combination of swimming speed, frequency, and amplitude at which cost of transport is minimal. The swimming speed at this global cost-of-transport minimum, however, is rather low. If fish swim at a sub-optimal swimming speed, they can still optimise their energetics by swimming with a frequency-amplitude combination that minimises cost of transport for that particular speed. Yet at sub-optimal speeds, costs of transport will never be as low as at the global optimum, which requires the fish to swim at the optimal speed using the combination of amplitude and frequency optimal for that speed. We hope that this explanation clarifies the difference between global and speed-specific optima.

[Point 35] 8) Why is amplitude normalized but frequency isn't?

Reply: Thank you for your query. By normalising amplitude, but not frequency, we follow standard disciplinary practices. Normalisation can be desirable when studying phenomena across scales, specifically in studies such as this one, it facilitates comparisons and analysis across multiple species that range widely in size. We hope that these explanations address the reviewer's question.